# Resolving GIA in response to modern and future ice loss at marine grounding lines in West Antarctica

5    Jeannette Xiu Wen Wan[1], Natalya Gomez[1], Konstantin Latychev[2], Holly Han[1,3]

[1]McGill University, Department of Earth and Planetary Sciences, Montreal, Canada
[2]Harvard University, Department of Earth and Planetary Sciences, Cambridge, Massachusetts
[3]Los Alamos National Laboratory, Fluid Dynamics and Solid Mechanics Group, Los Alamos, USA

10    *Correspondence to*: Natalya Gomez (natalya.gomez@mcgill.ca)

**Abstract.** Accurate glacial isostatic adjustment (GIA) modelling in the cryosphere is required for interpreting satellite, geophysical and geological records and to assess the feedbacks of Earth deformation and sea-level change on marine ice-sheet grounding lines. GIA modelling in areas of active ice loss in West Antarctica is particularly challenging because the ice is underlain by laterally varying mantle viscosities that are up to several orders of magnitude lower than the global average, leading to a faster and more localized response of the solid Earth to ongoing and future ice sheet retreat and necessitating GIA models that incorporate 3-D viscoelastic Earth structure. Improvements to GIA models allow for computation of the viscoelastic response of the Earth to surface ice loading at sub-kilometre resolution, and ice-sheet models and observational products now provide the inputs to GIA models at comparably unprecedented detail. However, the resolution required to accurately capture GIA in models remains poorly understood, and high-resolution calculations come at heavy computational expense. We adopt a 3-D GIA model with a range of Earth structure models based on recent seismic tomography and geodetic data to perform a comprehensive analysis of the influence of grid resolution on predictions of GIA in the Amundsen Sea Embayment (ASE) in West Antarctica. Through idealized sensitivity testing down to sub-kilometer resolution with spatially isolated ice loading changes, we find that a grid resolution of $\sim\frac{1}{3}$ of the radius of the load or higher is required to accurately capture the elastic response of the Earth. However, when we consider more realistic, spatially coherent ice loss scenarios based on modern observational records and future ice sheet model projections and adopt a viscoelastic Earth, we find that predicted deformation and sea-level change along the grounding line converge to within 5% with grid resolutions of 7.5 km or higher, and to within 2% for grid resolutions of 3.75 km and higher, even when the input ice model is on a 1 km grid. Furthermore, we show that low mantle viscosities beneath the ASE lead to viscous deformation that contributes to the instrumental record on decadal timescales and equals or dominates over elastic effects by the end of the 21st century. Our findings suggest that for the range of resolutions of 1.9-15 km that we considered, the error due to adopting a coarser grid in this region is negligible compared to the effect of neglecting viscous effects and the uncertainty in the adopted mantle viscosity structure.

## 1 Introduction

Changes in sea level in response to ice-mass loss are spatially variable because of glacial isostatic adjustment (GIA), the deformational, gravitational, and rotational response of the Earth to changes in surface ice and water distribution. The response of the bedrock is viscoelastic, beginning with an instantaneous elastic response of the solid planet's lithosphere and mantle, and transitioning to a longer timescale viscous relaxation of the mantle towards isostatic equilibrium. GIA models produce predictions of changes in the height of the sea surface equipotential and solid Earth surface (i.e. sea-level changes) in response to surface ice and water loading changes, which are in turn used to interpret satellite, geophysical and geological records and serve as input to models of ice-sheet dynamics.

Accurate GIA modelling is required to constrain the sea level and solid Earth feedbacks on ice dynamics in the coming centuries, especially along unstable marine-grounded ice fronts in Antarctica where bedrock uplift and gravitational drawdown of the sea surface due to ice loss act as a negative feedback to stabilise the retreat of marine-grounded ice-sheet grounding lines (e.g. Gomez et al., 2010; 2013; 2015; De Boer et al., 2014; Konrad et al., 2015; Larour et al., 2019). Furthermore, the Earth's response to past and modern ice cover changes makes a significant contribution to satellite records of modern mass changes in marine sectors of West Antarctica that are actively experiencing ice loss (e.g. King et al., 2012; the IMBIE team, 2018).

GIA modelling in Antarctica is complicated by the fact that the continent is characterised by strong lateral variability in lithospheric thickness and upper mantle viscosity, with low viscosities in the west and high viscosities in the east (e.g. An et al., 2015a; Heeszel et al., 2016; Lloyd et al., 2020). In particular, the low-viscosity mantle and thin lithosphere observed under the West Antarctic Ice Sheet (WAIS) identified from increasingly resolved seismic tomography and geodetic and geologic constraints (Ritzwoller et al., 2001; Morelli and Danesi, 2004; Kaufmann et al., 2005; Nield et al., 2014; Heeszel et al., 2016; Barletta et al., 2018; Shen et al., 2018; Lloyd et al., 2020) leads to a more spatially localised (short wavelength) and faster viscoelastic response to surface loading than cratonic regions covered by Late Pleistocene ice sheets (e.g. Hay et al., 2017; Powell et al., 2020). Over West Antarctica, upper mantle viscosities are thought to vary by several orders of magnitude over short spatial scales ( ~100s of km) reaching as low as $10^{18}$ Pa s in the Amundsen Sea Embayment (ASE) beneath areas of active marine ice loss (e.g. Nield et al., 2014; Barletta et al., 2018). This implies that viscous effects due to 20[th] century and more recent ice loss will become significant on annual to decadal timescales and accelerate during the timeframe of instrumental records (Barletta et al., 2018; Powell et al., 2020). Viscous deformation due to ongoing ice loss also has the potential to influence ice-sheet grounding lines in the coming centuries (Gomez et al., 2015; Kachuck et al., 2020; Coulon et al., 2021) but has not been included in recent high resolution coupled ice sheet – sea level model projections (Larour et al., 2019).

To accurately capture the timing and wavelength of GIA effects across Antarctica, models must be capable of both accounting for 3-D Earth structure (i.e., 3-D GIA models such as Latychev et al., 2005 or van der Wal et al., 2015), and be of sufficient spatiotemporal resolution to capture the geometry of grounded ice cover (e.g. Han et al., 2022). Commonly, GIA, ice-sheet and coupled ice-sheet – GIA modelling (e.g. Gomez et al., 2015; Konrad et al. 2015) studies that consider modern and future ice-sheet changes have been performed with only 1-D (radially varying) Earth structure models (e.g., Kendall et al., 2005; Spada and Stocchi, 2007; Adhikhari et al., 2016), or with coarse spatial resolutions of $> 20$ km due to the computational expense. GIA models capable of km- to sub-km-resolution have also been developed (e.g. the 1-D GIA model by Adhikhari et al., 2016; the 3-D GIA model by Latychev, et al. 2005 with updates described in supplementary materials of Gomez et al., 2018). For computational efficiency, some of these models implement regional grid refinement techniques which allow for a higher resolution along ice retreat margins. Alongside this, improvements in ice-sheet models (e.g. Nowicki et al., 2016; Seroussi et al., 2020; DeConto et al., 2021) and observational products (e.g. Studinger, 2014; Bamber et al., 2020; Smith et al., 2020; Morlinghem et al., 2020) have been made to provide similarly high-resolution (km- to sub-km-) ice thickness and bedrock topography datasets that serve as input to GIA models. These advancements together allow for GIA models to capture short-wavelength bedrock deformation and input ice loading changes at unprecedented detail, but at still heavy computational expense, particularly for global 3-D GIA models.

It is well-established that dynamic ice-sheet models are sensitive to the chosen grid resolution (e.g. Durand et al., 2009; Van den Berg & Van de Wal, 2006), requiring km to sub-km resolution to accurately represent ice dynamics and grounding-line migration in some applications (e.g. Gladstone et al., 2012; Pattyn et al., 2013; Cornford et al., 2016). It has also been suggested that as fine as 1 km resolution bedrock topography may be needed to capture the influence of fine-scale topographic features on the ice-sheet evolution (Durand et al., 2009) and high resolution may also be needed to represent some embayment walls and pinning points that act to slow down retreat (e.g. Favier et al., 2012; Joughin et al., 2014; Berger et al., 2016).

While topographic features themselves can be very fine scale, the changes in bedrock elevation and sea level in response to ice cover changes tend to be longer wavelength, and the corresponding spatial resolution required to accurately resolve these changes in GIA models and their influence on ice dynamics remains poorly understood. Larour et al. (2019) suggested that kilometer-scale resolution may be required to capture the elastic component of deformation in response to ice loss. However, the idealized tests they performed considered an isolated, and increasingly localized load, and their conclusion may not hold for more realistic, spatially coherent ice loss scenarios. Furthermore, their model did not include viscous deformation in response to ongoing ice loss during the simulation, or account for lateral variations in Earth structure. There have been limited studies assessing the length scale of realistic viscoelastic bedrock response in the rheologically complex region beneath the WAIS, though a recent high-resolution bedrock deformation modelling study by Zwinger et al. (2020) suggests a convergence in modelled viscoelastic deformation at resolutions finer than 5 km. The broad spatial nature of bedrock deformation and the

spatially coherent nature of ice-sheet retreat, which becomes less localised over longer timescales, suggest that sub-km to km

grid resolution, which comes at great computational cost, may not be necessary for accurate GIA model calculations.

The aim of this study is to assess the sensitivity of GIA-model predictions of Earth deformation and sea-level change in response to modern and future ice loss to spatial resolution in the rheologically complex, marine sectors of the WAIS. We build a 3-D viscosity model based on the most recent Antarctic-wide seismic tomography model (Lloyd et al., 2020) to serve

as input to a 3-D finite volume, global GIA model (Latychev et al., 2005) to assess the performance of 3-D GIA model predictions across surface grid resolutions of 1.9-15 km. We repeat calculations with a range of Earth models, considering the contribution from elastic and viscous deformation separately. We focus on the response to observed modern ice loss over the last two decades (Shepherd et al., 2019) and projected future ice-sheet retreat in the coming century (Golledge et al., 2019; DeConto et al., 2021) in the Amundsen Sea Embayment of West Antarctica. Our study is motivated by the following questions:

What 3-D grid resolution is necessary to adequately capture the elastic and viscous deformation and sea-level changes in response to ice loading changes? How significant is the effect of grid resolution compared to sources of uncertainty and simplifications made in some previous modelling, in particular the neglect of viscous deformation?

## 2 Methods

To investigate the influence of GIA model grid resolution, we first conduct idealized load sensitivity tests over a range of surface grid resolutions from 7.5 to 0.5 km, for the instantaneous removal of cylindrical loads from 0.5 to 16 km in radius (Section 3). We then widen our "aperture" to assess the model grid resolution required to accurately capture GIA due to modern ice cover changes from satellite observations (Shepherd et al., 2019) and projected ice loss from ice-sheet models (Golledge et al., 2019; DeConto et al., 2021) in the rapidly evolving Amundsen Sea Embayment of West Antarctica (Section 4). We

choose to locate our study region (light blue square in Fig. 1a) on the ASE both because of the ongoing ice loss and vulnerability of marine-based ice sheets to large-scale retreat in the region, and because the region is characterized by low upper mantle viscosities and a thin lithosphere (e.g. Barletta et al., 2018), making the ice there sensitive to solid Earth and sea-level feedbacks. In the ASE, the Pine Island and Thwaites Glaciers together contributed 95 Gt/year of the 159 ± 8 Gt/year total WAIS mass flux in 2017 (Rignot et al., 2019), with studies estimating that collapse of Thwaites Glacier is already underway

(Joughin et al., 2014). Accurate GIA model predictions are critical to assess rates of future ice-sheet retreat and associated sea-level changes making it an ideal location to study the effects of grid resolution on modelled GIA. To represent the radially and laterally variable Earth rheology in this region, we use a viscoelastic Earth rheology and a range of 3-D viscosity structure models in Antarctica derived from seismic tomography (An et al., 2015a; Heeszel et al., 2016; Lloyd et al., 2020). We adopt a range of 3-D Earth model grids with surface resolutions from 1.9-15 km and compare results to first assess the resolution

required to capture the elastic deformation associated with ice loss. We then repeat these experiments with viscoelastic effects

and compare results to the elastic calculations to assess the contribution of viscous effects to modern and future sea level, and the model resolution required to capture these effects. In the sections that follow, we describe the adopted 3-D GIA model, computational grids, Earth rheological models, and adopted modern and future ice loss scenarios.

## 2.1 3-D GIA Model

We perform all our simulations with a global, 3-D, finite volume GIA model (Latychev et al., 2005) capable of regional grid refinement (Gomez et al., 2018). The model solves the sea-level equation (Kendall et al., 2005) with time-varying shorelines over a 3-D, spherical, tetrahedral grid defined globally from the surface to the core-mantle boundary (CMB), allowing us to resolve the laterally and radially varying Earth structure that is a strong feature in Antarctica. We also adopt this 3-D GIA model because it is capable of regional grid refinement to achieve regional resolution at sub-km scale in regions of interest
within a lower resolution globe (Section 2.2). The model computes deformation of the solid Earth and gravitationally self-consistent changes in position of the sea surface equipotential in response to applied ice loading changes accounting for the effects of perturbation in the Earth's rotation and solid surface assuming an elastically compressible Maxwell viscoelastic rheology. Note that this henceforth called "GIA model" may also equivalently be referred to as a "sea-level model" in some literature.


The GIA model requires two main inputs, a 3-D Earth model of viscoelastic rheological properties and a time series of ice thickness changes (where the location of grounded ice may be prescribed or computed within the model if the ice thickness data includes floating ice). These components are described in the following sections. The model also requires global ice-free topography as an initial boundary condition, including the elevation of the bedrock beneath the applied ice load. Outside of
Antarctica, etopo2 from NOAA (NOAA National Geophysical Data Centre, 2006) is used in all experiments with a realistic loading scenario, and the Antarctic bedrock elevation for each of the experiments is described below. Note that we adopt a standalone GIA model throughout this study with the purpose to inform the setup of coupled ice sheet-GIA modelling studies, but we do not model the feedback of GIA effects on ice-sheet dynamics explicitly. However, other studies (e.g. Gomez et al., 2015; Kachuck et al., 2020) suggest that the scale of differences in solid Earth deformation and sea-level change simulated at
different resolutions and adopting different Earth structure models here will be large enough to alter the timing and magnitude of grounding-line migration in a coupled modelling context.

## 2.2 Computational model grid: regional grid refinement

To compute GIA model predictions, we construct model grids of various surface resolutions (Fig. 1) with the regional grid refinement process detailed in the supplementary material of Gomez et al. (2018). Grid refinement is achieved by incrementally
bisecting grid edges over a selected 3-D region to achieve a desired resolution, and a final smoothing operation along the region boundary to ensure a well-behaved transition. We perform calculations on a base grid with a global surface resolution of 15 km, which consists of 20 million nodes and 70 radial layers between the core-mantle boundary and the Earth's surface.

The radial layers of the grid are defined to respect the unconformities in material properties of the radially varying (1-D) seismic reference model STW105 (Kustowski et al., 2008), with the shallowest layers at 12, 25 and 43 km depth. Eight regionally refined grids are constructed from this base grid: five for the idealized load sensitivity tests at 7.5, 3.75, 1.75, 1 to 0.5 km surface grid resolution over a minimum 0.3-degree radius region around the test load used in Section 3, and three for the more realistic calculations from observed modern and future model projected ice loss in Section 4 at 7.5, 3.75 and 1.9 km surface resolution in incrementally smaller series of nested 3-D regions converging over the ASE (Fig. 1). Results in Section 4 are also computed on the base grid of 15 km resolution. The highest-resolution 1.9 km grid over the ASE has ~ 29 million nodes, which takes ~ 65 CPU hours to run per time step on a high-performance computing cluster. As our study focusses on surface grid resolution, the 3-D grid refinement region is limited to the surface down to ~10 km depth where the grid is bisected incrementally both horizontally and vertically. Testing with deeper grid refinement during experiment design process indicated that this was sufficient, and our results indicate that km-scale model resolution is only needed at the surface to accurately capture the geometry of surface loading. For consistent comparison of results from various grid resolutions, all results from Section 4 are interpolated onto the same 221x221 node grid evenly spaced in distance spanning the study region.

## 2.3 Earth rheological model

The spatial pattern and amplitude of surface deformation in response to ice loading are dependent on the underlying Earth structure. For the idealized sensitivity tests in Section 3, we adopt a purely elastic Earth model with a 1-D elastic and density structure based on Preliminary Reference Earth Model (Dziewonski and Anderson, 1981). In Section 4, we move to a set of realistic simulations using observed or modelled AIS ice loading, adopting 3-D viscoelastic Earth models with a range of viscosity structures constrained by seismic tomography (An et al., 2015a,b; Lloyd et al., 2020) and informed by GNSS-inferences of local mantle structure. The elastic and density structure for these models are based on seismic reference model STW105 (Kustowski et al., 2008). Laterally varying lithospheric thickness (Fig. S1d) in all simulations is a composite of a regional lithospheric thickness model by An et al. (2015a) over Antarctica, and a global lithospheric thickness model by Conrad and Lithgow-Bertelloni (2006) everywhere else. Over Antarctica, lithospheric thickness is scaled to have an average of 96 km, resulting in a minimum of 40 km, as was done in Hay et al. (2017).

To establish lateral variations in mantle viscosity, we follow Latychev et al. (2005) and Austermann et al. (2013) by sequentially converting relative variations in isotropic shear wave velocity to density, temperature and eventually viscosity. The latter step admits a free scaling factor to scale temperature changes inside an exponential term. It follows from Ivins and Sammis (1995) and Kaufmann et al. (2005) Equation 8, that this free scaling factor can be tabulated under certain assumptions and appears to be a slowly decreasing in magnitude with depth (see Supplementary Section S2 for details). We chose to adopt a free scaling factor because, generally, a different scaling is required to account for the amplitude differences between global and regional seismic models while constructing a composite model.

To address the substantial uncertainty in Earth structure, we repeat our simulations with three different 3-D viscosity models: EM1_L (Figs. 1c, d, S1), EM1_M, and EM2. EM1_M and EM1_L adopt the latest high-resolution 25 km Antarctic seismic tomography model by Lloyd et al. (2020) (ANT-20) in the region south of 45°S and extending from the surface down to the transition zone, and S362ANI (Kustowski et al., 2008) for the rest of the globe. ANT-20 is provided on a 25 km grid at the surface, with distance between the points decreasing with depth, and provided at depth slices in 5 km intervals from 0 to 800 km depth. S362ANI is a 3-D global anisotropic shear wave velocity model for the whole mantle, extending from 25 km depth to the core-mantle-boundary defined at 2891 km depth, provided at a 2° lateral grid resolution at the surface. We note, however, that the spacing of the seismic model grid is distinct from the scale of the Earth structure variations captured by the model. While quantitatively assessing the latter remains an outstanding challenge in seismic tomography, we expect the resolution of the ANT-20 model to be $\mathcal{O}(>100$ km) in the upper mantle and coarser at greater depths, and thus well represented by the GIA model grid. The two variations EM1_M and EM1_L were scaled to represent a moderate range of viscosities across Antarctica that match regional averages, and a lower-viscosity endmember adjusted to match GPS-derived inferences of minimum viscosity beneath WAIS (Nield et al., 2014; Zhao et al., 2017; Barletta et al., 2018).

The viscosity variations in EM1_M are more moderate with scaling factors of 0.0263 $K^{-1}$ for ANT-20 and 0.035 $K^{-1}$ for S362ANI, both close to the value arising from Kaufmann et al. (2005) (see Supplementary Section S2). These viscosity variations are superimposed on a 1-D viscosity profile of $5 \times 10^{20}$ Pa s in the upper mantle and $5 \times 10^{21}$ Pa s in the lower mantle typical in most GIA-based inferences of mantle viscosity (e.g., Mitrovica and Forte, 2004). As noted, the viscosity variations in EM1_L (Figs. 1c, d, S1) are of higher amplitude with lower viscosities under WAIS and the ASE. EM1_L adopts larger scaling factors of 0.033 $K^{-1}$ for ANT-20 and 0.04 $K^{-1}$ for S362ANI and an accompanying 1-D viscosity profile of $1 \times 10^{20}$ Pa s in the upper mantle, which is aligned with Lambeck et al., (2014) estimates of 1-D upper mantle viscosity using far-field sea-level proxy records, and $5 \times 10^{21}$ Pa s in the lower mantle. The larger scaling factors applied in EM1_L were calibrated to best reflect the absolute upper mantle viscosity estimates from GPS bedrock uplift rates at three locations: $\sim 6 \times 10^{17}$ to $2 \times 10^{18}$ Pa s at the northern Antarctic Peninsula (Nield et al., 2014), $\sim 2 \times 10^{19}$ to $2 \times 10^{20}$ Pa s at the Fleming Glacier in the central Antarctic Peninsula (Zhao et al., 2017), and $\sim 2.5 \times 10^{18}$ to $4 \times 10^{19}$ Pa s at the Amundsen Sea Embayment (Barletta et al., 2018). Figures 1c, d and S1 shows the resulting low viscosity earth model structure (EM1_L), which has the lowest viscosity at Marie Byrd land of $\sim 9 \times 10^{17}$ to $7 \times 10^{18}$ Pa s in the upper mantle.

Lastly, the EM2 model, also adopted in Hay et al. (2017), Gomez et al. (2018) and Powell et al. (2020), is a combination of three seismic models: S40RTS (Ritsema et al., 2011) globally, a model by An et al. (2015a) in East Antarctica and the Antarctic Peninsula, and the model by Heeszel et al (2016) for West Antarctica. The full construction of EM2 is detailed in Hay et al. (2017).

## 2.4 Ice model and topography

We consider three ice melt scenarios with resolution ranging from 1- to 5 km in the ASE. The total ice thickness change from start to end of each scenario is shown in Figs. 2a-c.

For observations of modern ice loss, we adopted a reconstruction we term ICE-SH from Shepherd et al. (2019) of surface elevation change ($\Delta h$) from 25 years (1992-2017) of multi-mission satellite altimetry data resolved over a 5 km grid at 5-year intervals. We treat $\Delta h$ as a proxy for ice thickness change (Carrivick at al., 2019) and apply the Bedmap2 (Fretwell et al., 2013)

grounded ice mask and saturate ice thickness change > 20 m/yr to control against spurious data points. Initial ice thickness and bedrock topography in Antarctica is given by Bedmap2. Observations of ice surface elevation changes in Antarctica are continuously improving in resolution, and currently range from metre-scale resolution over short observational tracks (e.g. Studinger et al., 2014), to sub-km to km-scale resolution at the regional scale (e.g. Bamber et al., 2020), to ~ 5 to 35 km from radar and laser satellite altimetry derived records over the whole Antarctic (e.g. Martin Español et al., 2016; Schröder et al.,

2019; Shepherd et al., 2019; Smith et al., 2020). ICE-SH was selected from the available observational datasets due to the 5 km resolution being the highest of its class of available satellite altimetry derived-records providing decadal time span surface elevation change records covering the whole Antarctic.

For Antarctic evolution over the next century, we apply modelled ice thickness changes from two Antarctic-wide ice-sheet

model projections: (1) ICE-GOL which predicts AIS evolution under RCP 8.5 and including meltwater feedbacks from 2000 to 2100 at 5 km resolution over 5-year intervals (Golledge et al., 2019), and (2) ICE-RD which predicts AIS evolution from 1950 to 2100 at 10 km resolution over the whole AIS with a nested 1 km resolution simulation over ASE at annual resolution (Extended Data Fig. 5 from DeConto et al., 2021). For ICE-RD, we ran simulations at yearly intervals from 1950-2100 but the interval from 1950-2000 is a period of ice model initialization and we therefore focus on the period between 2000 and 2100 in

our results. We also take initial Antarctic bedrock topography from the ice models. Further information of each model can be found in the corresponding references. In selecting these scenarios, the goal is to provide a representative sample of spatially coherent ice-sheet retreat scenarios at high resolution from the literature, rather than to capture all possible projected future ice loss scenarios.

When inputting a given ice load into the 3-D GIA model, the load mapper algorithm interpolates via a non-linear scheme, the equivalent load acting on each triangular area in the computational grid. Subsequently, an equivalent of 1/3 of the share of the load falling on each triangle grid area incident on the node is summed onto the loaded computational grid node. Within the computational grid triangle area, the load is assumed to be a linear function in triangular coordinates.

## 3 Idealized experiments: sensitivity of elastic uplift predictions to grid resolution

Our main goal in this analysis is to assess the relationship between grid resolution and GIA model predictions, and to identify, for a given load dimension, the grid resolution required to accurately model the associated elastic Earth deformation component of the sea-level change. Realistic ice retreat has complex geometry, making it difficult to pinpoint the cause of inaccuracies due to resolution, which may be due to poor representation of the ice load, or numerical errors in representation of the response to Earth loading. To isolate the effect of changing grid resolution on sea-level predictions, we first perform a suite of idealized

sensitivity tests using the GIA model described in Section 2.1 to make predictions of the instantaneous elastic deformation from unloading of an isolated cylindrical ice load with differing surface grid resolutions that are iteratively bisected from 7.5 km down to 0.5 km. We chose to perform the test with short wavelength, spatially isolated ice-loading changes because these would be most poorly represented by a coarse grid compared to coherent ice loss over a broader area. Furthermore, these tests with idealized loads are less computationally costly and enable a systematic assessment reaching higher spatial resolution. In

total, 85 simulations were run using 17 ice cylinders of height 100 m and radii ranging from 0.5 to 16 km (0.5, 1, 2,…,15, 16), across five different grid surface resolutions: 0.5, 1, 1.75, 3.75 and 7.5 km.

The simulations are performed with the purely elastic 1-D Earth model (Sect. 2.3), and an idealized topography of 3800 m south of 24.5 °S and - 835 m everywhere else to reflect the 30:70 land to sea ratio on Earth (Fig. S2). A radially symmetric ice

sheet with steady-state Antarctic ice dome profile (Paterson and Colbeck, 1980) sits on top of this topography extending from the south pole to 69°S, with a maximum height of 3500 m. We also consistently place the centre of the cylindrical load on an arbitrary model grid node in the ASE (76°S 150°W). All results from Section 3 are plotted by interpolating onto the same 80 km box with 200 m resolution over the study region.

### 3.1 Idealized experiment results

Figure 3 summarises the error in the predicted elastic solid Earth deformation with varying load radius and grid resolution. Figure 3a shows the bedrock deformation along a transect from the centre of the load for ice cylinders of 2, 5 and 10 km radius, with maximum bedrock uplift predicted on the finest 0.5 km grid of 48, 108 and 190 mm, respectively. Figure 3b indicates whether the grid over- (blue) or under- (red) represents the mass of the load within the model (i.e. where the "Mass Factor" is less or greater than 1). Errors in the solid Earth deformation predictions, reported relative to the result for the finest resolution

0.5 km grid (yellow lines), are typically the largest at the load centre where they underestimate the magnitude of peak displacement. Although a coarser grid may either under- or over-estimate the mass of loading represented in our model (Fig. 3b), it will always dampen the magnitude (effective height) of the load by spreading the load area over a larger grid region. For example, a 5 km radius load will be represented by 3 grid nodes on a 7.5 km grid, resulting in an overestimated 11.25 km radius loading footprint. For certain radius and grid combinations, the wider load footprint on a coarser grid leads to another

zone of peak error occurring outside the grounded load edge (e.g., compare dashed black line to yellow line in Fig. 3a). Even further from the load, the magnitude of deformation decreases and the results from various grid resolutions begin to converge.

These sensitivity test results highlight that the accuracy of predictions depends on the placement of the edge of the load relative to grid nodes and find that the load will be best captured if its edge lies sufficiently close to a grid node (e.g., in Fig. 3b the 1.75 km grid more closely captures the mass of a 2 km radius load than the 3 km radius load). Finally, the grid is unable to resolve the load when the grid resolution is more than approximately three times the radius of the load. This is illustrated, e.g., in the black solid line in Fig. 3a, where unloading a 2 km radius load on a 7.5 km grid resulted in no deformation, whereas a 3 km radius load is captured with a 7.5 km grid.

To quantify grid-related error, we calculate the difference between a given simulation and the corresponding simulation with the finest (0.5 km) resolution. We plot the root mean square error (RMSE; Fig. 3c) as an absolute measure of error and the average and standard deviation of the absolute percentage error (Figs. 3d, e) as a relative measure of error, beneath and within 2 km of the loaded region. Fig. 3c shows that the magnitude of RMSE remains relatively constant. This RMSE remained between ~10-20 mm for a 3.75 km grid, and ~ 20-40 mm for a 7.5 km grid, for example. As the load radius increases, the magnitude of deformation increases as well. However, the error due to grid resolution becomes less significant relative to the total deformation (i.e. the percentage error in Figs. 3d, e decreases).

While the dependence of grid performance on load position relative to grid nodes complicates matters, in order to arrive at an approximate relationship between grid resolution and load size, we assume a linear relationship between the two, which allows us to estimate, for this GIA model, a threshold beyond which grid-related error becomes sufficiently low to no longer merit a further refinement in grid resolution (Figs. 3d, e). For example, in the cross-sectional view of deformation in Fig. 3a, the 10 km radius load deformation is equally well represented by grid resolutions between 0.5 to 1.75 km. Considering the average absolute percentage error (Figs. 3d, e), we found that a 1:3 ratio (represented visually on Figs. 3b-e in the form of a black dashed line) between grid resolution and load radius (1:6 ratio between grid resolution and load diameter) brings the error to < 7 ±3 % (where 3% represents one standard deviation of the absolute percentage error calculated within 2 km of the load region). Furthermore, the mass of the load is accurately captured with no more than ~7.5% error with this 1:3 ratio (Fig. 3b).

The results from this analysis of spatially isolated cylindrical loads provide a rough estimate of the magnitude of error one can expect from a given model resolution and loading scenario, and can serve as a guide for selecting the appropriate grid resolution for a given problem. For example, for an input load with significant isolated locations of ice loss ~ 3 km in radius (or a ~ 6 km in diameter), a grid resolution of 1 km should be adopted. However, these idealized cylindrical load experiments are unlikely to capture the sensitivity of deformation and sea-level predictions to grid resolution when realistic ice loss geometries are adopted. Such geometries are rarely characterized by spatially localized loads. Furthermore, these experiments capture only

elastic deformation and neglect viscous effects, which can be significant on short timescales in low viscosity zones of the West

Antarctic. In the following sections we explore how the dependence on grid resolution of GIA model predictions identified here changes when more realistic ice loss geometries and 3-D viscoelastic Earth structure are adopted.

## 4 Results with realistic modern and future ice loss in the Amundsen Sea

In this section, we consider the importance of grid resolution error for more realistic, spatially coherent modern and future ice loss scenarios. We begin with a consideration of the influence of grid resolution on predicted sea-level change in simulations

adopting a purely elastic Earth model in Section 4.1. Following this, we adopt 3-D viscoelastic Earth models to consider the contribution to sea-level change from viscous deformation. Throughout Section 4, grid resolution error is defined as departures of predicted sea-level changes from the finest resolution 1.9 km grid resolution result.

## 4.1 Influence of grid resolution on sea-level predictions with elastic deformation

Figures 2d-f show predicted sea-level change in the Amundsen Sea Embayment of West Antarctica adopting an elastic Earth model for three different ice retreat scenarios, performed on a grid resolution of 1.9 km: one observationally constrained from 25 years of satellite altimetry data from 1992 to 2017 (ICE-SH; Fig. 2a), and two projected from dynamic ice-sheet models for the coming century (ICE-GOL and ICE-RD; Figs. 2b-c). Sea-level fall is predicted in the entire study region in all scenarios associated with the combination of sea surface subsidence and elastic bedrock uplift due to ice loss – the latter being the

dominant signal. Earth rotational effects are included in the predictions but are negligible compared to the other effects in the vicinity of the ice loss. For the modern, the maximum sea-level fall from 1992 – 2017 reaches 0.68 m (Fig. 2d), while for future ice loss projections, the sea-level fall reaches up to 9.06 m and up to 12.8 m from 2000 – 2100 for ICE-GOL and ICE-RD respectively. Note that in addition to signal coming from local ice loss in the ASE, ice outside this region of interest also contributes a broad signal of smaller magnitude (see Supplementary Section S1).


To explore the resolution dependence of sea-level predictions that adopt an elastic Earth model, we repeat the calculations in Figs. 2d-f with a surface grid resolution ranging between 1.9 and 15 km. Figs. 4a-i, shows the difference between results for simulations performed at 1.9 km grid resolution relative to coarser resolutions. The coarser grid simulations tend to underestimate the magnitude of ice unloading and associated sea-level fall in most of the domain (red regions in Fig. 4).


The highest grid resolution error occurs at the periphery of the load within a few kilometres of the final grounding line position rather than at the location of maximum deformation (compare Fig. 4 to Fig. 2). This suggests that high resolution is necessary for better representation of the load at the grounding line, rather than for representation of the smoother response of the solid Earth – a finding consistent with the idealized experiments discussed in Section 3 (see Fig. 3b). For example, for the ICE-GOL

ice loss scenario, the greatest difference between 1.9 and 15 km grid simulations ranges occurs along the entire final grounding line (Fig. 4b), but the maximum sea-level fall of over 9 meters occurs only on a concentrated region ~ 2 km away from the grounding line (Fig. 2e).

The error decreases with increasing resolution, with minimal differences between the 1.9 km and 3.75 km grid resolutions. 360 The maximum absolute error in the case of a 15 km grid (i.e. the maximum difference between the 15 km and 1.9 km resolution cases) is 44 cm at 2100 in ICE-RD (Fig. 4c), 45 cm at 2100 in ICE-GOL (Fig. 4b) and 8.7 cm after 25 years of modern melt in ICE-SH (Fig. 4a). That is 3.4%, 5.0% and 12.8% of the peak elastic sea-level fall predicted at that time for each respective scenario. The errors are approximately an order of magnitude smaller when a grid resolution of 3.75 km is adopted: 9 cm for ICE-RD, 4 cm for ICE-GOL and 0.7 cm for ICE-SH, or 0.7%, 0.5% and 1.0% of the peak elastic sea-level fall respectively.


Since maximum grid resolution error is concentrated along the grounding line for elastic simulations, in Fig. 5 we explore how the error evolves during the ICE-RD simulation along a 10 km region bounding the grounding line. The error increases in absolute magnitude with increasing ice loss ("Error" in Fig. 5) but the relative error decreases across the same simulations ("Percentage Error" in Fig. 5). In the case of 15 / 7.5 / 3.75 km grid resolutions, the peak error is ~10 / 5 / 1 cm after 25 years 370 in the simulations, and ~50 / 15 / 5 cm after 150 years (whiskers in Fig. 5a top row). In contrast, the percentage, peaks at 20 / 6 / 1.5% of the signal at 25 years and drops to < 5 / < 2 / < 0.3% after 150 years (Fig. 5). This decrease in percent error with time reflects that the ice geometry changes become broader in wavelength and can therefore be resolved by a coarser grid compared to the more spatially isolated changes occurring earlier in the simulation. Given the resolution in modelled and observed ice loss and bedrock elevation in Antarctica (e.g. Morlinghem et al., 2020), we suggest that for most applications, 375 errors of less than 5% can be achieved with a 7.5 km grid, and errors of less than 2% with a 3.75 km grid.

## 4.2 Contribution of viscous Earth deformation to sea-level predictions

So far we have focussed on the resolution dependence of the contribution to sea-level change from elastic Earth deformation, as this has been a focus of recent literature (Larour et al., 2019). However, the Antarctic Ice Sheet is underlain by strongly laterally varying viscosity structure, and the Amundsen Sea region in particular is underlain by a low viscosity zone and thinned 380 lithosphere (e.g. Barletta et al., 2018; Lloyd et al., 2020). Viscous deformation associated with ongoing ice loss is neglected in Larour et al. (2019) but is expected to be significant on decadal to centennial timescales in this region. In Figs. 2g-i the calculations of sea-level change associated with the three ice loss scenarios shown in Figs. 2d-f are repeated with the 3-D viscoelastic Earth EM1_L described earlier. As with the elastic case, sea level falls beneath regions that experience ice loss in all three cases, but the magnitude of the sea-level fall is significantly larger than predictions based on an elastic Earth model 385 (compare Figs. 2g-i to Figs. 2d-f). In particular, peak sea-level fall in this case reaches -0.79 m over 25 years in the ICE-SH ice loss scenario, and -14.9 m and -29.1 m from 2000 to 2100 for ice loss scenarios ICE-GOL and ICE-RD, respectively. The latter (Fig. 2i) is more than double the sea-level fall calculated with the elastic Earth model (Fig. 2f).

Figure 6a-c shows the contribution of viscous Earth deformation to the sea-level predictions, calculated by taking the difference
between the full viscoelastic calculation shown in Figs. 2g-i and the calculation with an elastic Earth model shown in Figs. 2d-
f. Over the 25 year modern ice loss scenario (Fig. 6a), viscous effects contribute up to 12 cm, or 15% of the peak viscoelastic
prediction. In the future projections, within 100 years, the viscous contribution reaches up to 6.15 m of sea-level fall, or 41%
of the peak viscoelastic signal in predictions based on ICE-GOL and up to 17.7 m fall, or 61% of the peak viscoelastic signal
for ICE-RD, making viscous effects the dominant contributor over elastic effects in this latter case. For the future projections,
compared with the elastic signal (Figs. 3e-f), the zones of maximum viscous uplift and sea-level fall (i.e. zones of intense red
in Figs. 6b-c) are centered farther out beneath regions that experienced ice mass loss sooner in the simulation and have had
more time for viscous deformation to occur (Figs. 6a-c), but as we highlight below, substantial viscous deformation still occurs
along the current grounding line in the simulation. This is less evident in the modern because migration of the location of
maximum ice mass loss is minimal. Faint blue areas further from the region of ice retreat in Fig. 6b indicate a sea-level rise
due to peripheral bulge subsidence, a viscous process that results from the return flow of mantle material assuming viscous
incompressibility.

## 4.3 Influence of resolution on sea-level predictions with viscoelastic deformation

In Fig. 7, we repeat the assessment of grid resolution error in Fig. 4, but with a viscoelastic rheology based on the 3-D earth
model EM1_L. With the inclusion of viscous behaviour, the magnitude of the grid resolution error is similar to the elastic case
(compare the range of errors on Figs. 7 and 4) but the spatial pattern of the error becomes more complex. The maximum error
is no longer solely concentrated along the current grounding line since the solid Earth continues to respond viscously to the
poorly resolved loading changes along previous locations of the grounding line. This is particularly evident in the ICE-RD
simulations where the grounding line retreats across a large area. In this case, the grid-resolution error over the region of past
ice loss and grounding-line migration is equal to or larger than the error along the active grounding line (Figs. 7b,c). The error
increases during the simulation as viscous deformation builds, and suggests it also has a dependence on the 3-D viscosity
structure both due to lateral variation captured and the impact of a lower-than-average viscosity upper mantle in the WAIS,
which we explore briefly in Section 4.3.

Note that in the blue region of Fig. 7c, the sign of the error is not the same as in Fig. 7f because ice retreat is not continuous
within this particular region in ICE-RD. Specifically, between the years ~2020 to 2050 in ICE-RD, the grounding line in this
blue region stays relatively fixed, experiencing multiple episodes of localized ice retreat and re-advance (unloading and
loading). Situations of re-advance tend to occur at lateral scales < 15 km, such that these sub-grid scale movements were not
adequately captured with a coarser 15 km grid.

**4.4 Earth model uncertainty**

To investigate the influence of uncertainty in prescribed mantle viscosity structure, we compare simulations adopting five different Earth model configurations: a globally averaged 1-D viscoelastic Earth model, as used in EM1_M described in Section 2.3, a best fit 1-D viscoelastic Earth model for the WAIS described further in Section 4.5, and three 3-D mantle viscosity configurations derived from seismic tomography models (EM1_L; EM1_M; EM2; see Sect. 2.3). Figs. 6d-f show the difference in sea level predicted using the EM1_L and EM1_M models, which are based on the same underlying 3-D

seismic velocity models but with different viscosity scaling factors and 1-D viscosity profiles. EM1_L (shown in Figs. 6a-c and 2g-i) has the lowest viscosity upper mantle beneath the ASE. Red regions in Fig. 6d-f indicate locations of higher predicted sea-level fall due to lower mantle viscosity and a shorter timescale of viscous response in EM1_L. Differences reach up to 5 cm after 25-years with ICE-SH, and up to 2.3 m and 5.8 m between 2000 and 2100 for ice loss scenarios ICE-GOL and ICE-RD, respectively. The simulation with EM2, a 3-D mantle viscosity model built from a different seismic tomography dataset,

produced deformation with magnitudes intermediate to the simulations with EMI1_L and EM1_M. We note that the rheology for this area is uncertain, and our experiments do not comprehensively capture the full range of this uncertainty (see Whitehouse et al., 2019 for a more detailed discussion).

**4.5 Time evolution of influence of grid resolution and viscous effects**

    To compare the relative contributions of grid resolution and Earth model differences over time, we extract predicted sea-level

time series from all simulations with the ICE-RD ice loading scenario and elastic and viscoelastic Earth models at two locations in Fig. 8: (X) the region experiencing the largest viscous uplift by 2100 (blue star), and (Y) the location of maximum ice loss at the 2100 grounding line (red star). Note that the shaded grey region from 1950-2000 in Figs. 8b and c represents a time of hindcast spin up to the year 2000 in the ice-sheet model simulation (see DeConto et al., 2021) rather than a realistic representation of the ice cover changes in this region over this time period. In the case of the globally averaged 1-D viscoelastic

Earth model, the sea-level response is similar to the elastic case (Fig. 8, compare red and black lines) because the upper mantle viscosity is set to $5 \times 10^{20}$ Pa s and thus a much longer timescale is required for viscous effects to become significant. In all cases, the differences between either the elastic or 1-D viscoelastic simulations and any of the viscoelastic simulations adopting 3-D Earth structure is substantially larger than the difference between the purely elastic and any 1-D viscoelastic simulation. For example, the differences between simulations using EM1_L and the global average 1-D viscoelastic Earth model, reach

up to 16.5 m and 12.7 m at the sites shown in Figs. 8b and c, respectively, by 2100.

    To briefly explore the distinction between the impact on predictions of adding lateral variations in Earth structure and the impact of adopting a 1-D model with lower than global average mantle viscosity profile representative of the structure underlying the WAIS, we ran an additional simulation adopting a 1-D viscoelastic Earth model for West Antarctica derived in

Powell (2021). This model has a viscosity of $3.2 \times 10^{19}$ Pa s in the shallow upper mantle, $1.3 \times 10^{20}$ Pa s in the deep upper

mantle, 2.0 x $10^{20}$ Pa s in the transition zone, and 5 x $10^{21}$ Pa s in the lower mantle. The predicted sea-level changes with the lower viscosity WAIS 1-D Earth model shown by the purple lines in Fig. 8 lie closer to the 3-D Earth model results than the global average 1-D Earth model at sites X and Y. The degree to which the response of the laterally varying Earth structure in this region can be captured with a radially varying approximation is explored in more detail in Powell (2021) and Blank et al.
455    (2021).

Starting from 50 years into the simulation at the year 2000, the influence of grid resolution becomes smaller than the effect of adopting different Earth models (compare the differences between the dashed and solid lines to differences between different coloured lines in Fig. 8). Using the 1-D global average Earth model (Fig. 8 red lines), viscous effects start to emerge after 50
years and reach only 4% of the peak signal by the end of the simulation. Nevertheless this signal is more significant than the error incurred by using 15 km versus 1.9 km grid resolution by 2040. With a 3-D Earth rheology and low viscosities beneath the ASE, viscous effects are pronounced within decades in the simulation (blue and green lines in Fig. 8) and become larger than the difference in predictions based on 15 km and 1.9 km grid resolutions within 25-30 years and before substantial ice loss has occurred in the simulation.


Fig. 9 provides a more detailed picture along the grounding line of the contributions to differences in predicted sea level described in the preceding sections. We consider the impact of each factor on the predicted sea-level signal at the end of the simulations, plotting the distribution of differences between simulations across all grid points within 10 km of the final grounding line. In interpreting Fig. 9, note once again that in viscoelastic simulations, the region of maximum grid-resolution
error does not necessarily occur near the grounding line (e.g. Figs. 7c, f). To visualise the distribution, we plot a classic box-whisker diagram where the edges of the boxes represent (from left to right) the 25th percentile, median and 75th percentile of the distribution, whilst the whiskers represent the "minimum" (25th percentile minus 1.5 times the interquartile range) and "maximum" (75th percentiles plus 1.5 times the interquartile range), overlain by a density curve. A box-whisker plot was chosen to mitigate against the effect of outlier points, which we plot as hollow diamonds.


Note that the error due to grid resolution consistently has a unimodal distribution that peaks at ~0-1% near the grounding line across a range of durations from 25-years of ICE-SH, to 100 and 150 years (starting from 1950) of ICE-RD. The hollow diamonds show that a significant number of points are statistical outliers, which is likely due to the fact that predicted sea-level change is low in magnitude at some regions along the grounding line that experience less ice-loss, causing even a small
magnitude of error to contribute a large percentage error. Inclusive of these outliers, the range of percent error due to a 15 km grid peaks at ~20% after 25-years in the ICE-SH simulation, but decreases to less than ± 8% by 100-years of the ICE-RD simulation as the magnitude of predicted sea-level fall becomes larger across the entire region and the spread of outlier points diminishes. We conclude that the range denoted by the box-whisker plot likely provides a more accurate assessment of the

error contributed by each factor in Fig. 9 in zones of active grounding-line migration. However, in the following paragraphs we continue to describe the range of errors inclusive of outlier points so as to not under-estimate the possible spread.

Our results indicate that the difference due to the choice of adopted Earth model equals and, in most cases, exceeds the size of the error due to grid resolution near the grounding line by the end of all our simulations. Over the 25-year modern observed ice loss scenario, the difference in predictions associated with Earth model configuration between results using EM1_L and EM1_M lies between ~2-10% within 10 km of the grounding line, which is within the range of error due to insufficient grid resolution in a viscoelastic simulation. The latter ranges from ± ~20% with a 15 km grid, ± 6% with a 7.5 km grid, and < 3% with a 3.75 km grid at all grid points (range of the diamonds outside the shaded distributions in Fig. 9a, though noting the discussion above, the percent error is substantially smaller than these end member values at most points). However, with more ice loss over longer timescales, the difference due to adopted Earth model far exceeds the grid resolution error (Figs 9b, c).

If we look beyond the grounding line and consider the difference in predicted sea level between different adopted 3-D viscosity models in the entire study region (difference in results with EM1_L and EM1_M, plotted in Fig. 6 d-f at 2100), the lower viscosity model results in additional viscous deformation that is up to 10.2 %, 21.4% and 20.9 % of the total viscoelastic signal predicted with EM1_L after 25-years of ICE-SH, 100-years of ICE-RD and 150-years of ICE-RD respectively. In all cases, the error due to neglecting viscous effects all together far surpasses the error due to grid resolution with a 15 km grid (compare the bottom rows to top 3 rows of Figs 9b and c). For example, the lower viscosity EM1_L model results in an additional viscous deformation that is up to 23.8 %, 58.9 %, 62.4 % more than the predictions with an elastic Earth model near the grounding line after 25-years of ICE-SH, and 100-years of ICE-RD and 150-years of ICE-RD respectively.

## 5 Discussion

Our study provides an assessment of the model grid resolution needed to capture decadal to centennial-scale Earth deformation and sea-level change in the vicinity of active ice loss. We targeted the ASE in West Antarctica as our study location as it is a region with ongoing and projected marine ice-sheet retreat and where low mantle viscosity and thin lithosphere result in a rapid and localised solid Earth response to ice loading. Whilst we focus on the ASE, since the resolution error is primarily associated with the representation of the ice load, our general conclusions on resolution requirements and results of the sensitivity experiments can be applied to any area of active, localised ice loading, for example in other parts of Antarctica, in Greenland or in the vicinity of smaller glaciers. We adopted a 3-D GIA model to accurately capture the viscoelastic response at high resolution, including the complexity introduced by laterally varying Earth rheology in the region. Accurate assessments of solid Earth deformation from past and present ice evolution are important for constraining the negative sea level – solid Earth feedback on ice-sheet retreat, and more accurate interpretation of geophysical observables. For the former, our study focusses on the sensitivity of sea-level predictions along the ice-sheet grounding line where this feedback occurs in Fig. 9.

## 5.1 Influence of grid resolution

For our suite of simulations with elastic and viscoelastic Earth models, modern and 21[st] century ice loss scenarios, and surface grid resolution ranging between 15 and 1.9 km, we found that improvements in the accuracy of GIA model predictions with increasing grid resolution was limited, remaining within centimetres to decimetres at the grounding line. Furthermore, our results converged at higher resolutions, with errors from a 3.75 km grid resolution reaching at most 6 cm within 10 km of the grounding line in all simulations, even when the input ice-sheet model results were available at 1 km resolution in the case of ICE-RD. The error introduced in assuming an elastic Earth model and neglecting viscous deformation in the ASE builds to an order of magnitude or more larger than the grid resolution error within three to four decades, and up to tens of meters by the end of the century. In addition, predictions adopting different 3-D Earth models that reflect the uncertainty in viscoelastic Earth structure in the region diverge by up to 1 meter within 50 years and upwards of 2-3 meters after 100 years in the simulation.

For coupled ice-sheet – GIA model applications, our results suggest that adopting high resolution in the ice-sheet model does not require a similarly high-resolution GIA model. In our simulations, a 3.75 km grid was sufficient to bring errors relative to the finest resolution simulation to < 2% along the grounding line for all scenarios (Fig. 9). Furthermore, this percentage decreased over time in our simulations, and would continue to decrease in multi-century and millennial simulations as the magnitude of viscous deformation and the scale of the ice loss continue to grow. While bedrock topography has smaller scale features (Morlinghem et al., 2020), our results suggest that the changes in topography will be less localized and may be computed at lower resolution relative to the ice-sheet dynamics and then interpolated and added to the initial topography on the higher resolution ice-sheet model grid, as is done in, e.g., Gomez et al. (2015) and DeConto et al. (2021).

Our results showing that the location of maximum error consistently lies along the grounding line for elastic Earth model simulations (Fig. 4a-c) suggest that the error due to coarse model resolution is predominantly a result of poor representation of surface ice cover changes rather than representation of the smoother response of the solid Earth. For the latter, we would expect the error to occur instead at the location of maximum sea-level response rather than in the vicinity of the edge of the load (compare differing spatial patterns in Fig. 2e, f to 4b, c). When the viscous response is incorporated, the time-evolving nature of viscous deformation leads to an additional peak in grid resolution error at locations predominantly downstream of the grounding line due to inaccurate representation of past loading (Fig. 7). This additional zone of error will not affect active ice-sheet grounding lines, though it may be important for interpretation of modern records or lead to re-grounding of an ice shelf. Note that while the spatial pattern of the error differs, the magnitude of the error due to grid resolution was similar across both elastic and viscoelastic simulations.

Our findings on the size and source of resolution error are in contrast to recent work by Larour et al., (2019) who suggested that kilometre resolution was required to capture elastic deformation. This discrepancy may be due in part to Larour et al.

(2019) considering only point loads in their idealized resolution experiment, while our conclusions are based on more realistic, spatially coherent ice loss scenarios. Differences may also arise due to the nature of the computational grid and processing of inputs (see Section 5.3) which should be explored in more detail in future GIA model inter-comparison efforts. Nevertheless, our predictions based on an elastic Earth model converge to theirs for more spatially broad loads.

One possible limitation in this study is that we do not reach sub-km grid resolution in our GIA model, and our highest resolution ice model is 1 km. In sensitivity tests with idealized loading scenarios in Section 3 we adopted a grid resolution as low as 0.5 km grid and found that a minimum 1:3 ratio between grid resolution and load radius was required for the error in predicted deformation to remain below $7 \pm 3$ ($\sigma$) % along the grounding line, suggesting that a 3.75km grid would be unable to capture a spatially isolated, < 1 km radius ice unloading event. That we did not see significant error at this resolution in the realistic simulations indicates that the ice cover changes are spatially coherent and there are no significant spatially isolated ice unloading events (i.e. no ice thickness changes occurring over only a few grid points) predicted in the 1 km resolution ICE-RD ice model simulation (Fig. 5).

To further investigate if short-wavelength, spatially isolated ice loss scenarios exist over Antarctica, we assessed the surface elevation change observables from 40 and 25 years of multi-mission satellite altimetry data by Schröder et al., (2019) and Shepherd et al. (2019) respectively, and 15 years of airborne laser altimetry from Operation IceBridge (OIB ATM L4; Studinger, 2014 (Updated 2018)). While a more detailed investigation is merited, in our initial analysis of these datasets we noted that spatially isolated ice loss events have a lower magnitude, only persist over short timescales, and found no evidence of high magnitude, short-wavelength ice loss occurring with spatial scales < 5 km. Thus, we expect that spatially isolated ice unloading occurs rarely and will not have a significant impact on the overall accuracy of GIA model results in a given region. Nonetheless, with improving observational products and ice-sheet model resolutions, we expect to obtain regional-scale ice loading grids of sub-km resolution that may warrant further study with a sub-km GIA model grid (e.g. Durkin et al., 2020).

**5.2 Influence of viscous deformation and Earth model uncertainty**

Within decades in the ASE, viscous deformation is a significant contributor to the GIA signal (Hay et al., 2017; Barletta et al., 2018; Powell et al., 2020; Kachuck et al., 2020). The GIA response can be decomposed into the following: perturbation of the sea surface equipotential, elastic deformation and viscous deformation of the solid Earth surface. Previous studies have isolated and assessed the relative importance of each of these contributions on ice-sheet dynamics. Over decadal to centennial timescales, Larour et al. (2019) confirmed that purely elastic deformation was more significant than the sea surface perturbation on continental-scales, while Kachuck et al. (2020) highlighted that viscous deformation can contribute dominantly. In this study, we have confirmed that viscous deformation effects are significant within decades, particularly in the low viscosity region of the ASE, where the viscous component of deformation can reach multi-metre scales by the end of the century. This

body of work highlights the importance of incorporating viscous behaviour in GIA modelling applications in regions of low mantle viscosity.

Complicating efforts to accurately characterise viscous deformation is the uncertainty in Earth's viscosity structure. The timescale of viscous solid Earth deformation on ice-sheet dynamics is strongly dependent on the assumed Earth rheology. The global average mantle viscosity of $\sim 10^{21}$ Pa s (Forte and Mitrovica, 1996) corresponds to response times from centuries to millennia, whilst recent seismic (Lloyd et al., 2020) and GPS observations suggest an upper mantle viscosity under the ASE as low as $\sim 10^{18}$ Pa s (Barletta et al., 2018). Rapid viscous uplift response was similarly identified in Kachuck et al. (2020), who a 2-D GIA model of mantle viscoelastic deformation and found that sea-level fall associated with viscoelastic mantle deformation led to a 30% reduction in modelled ice sheet volume loss by 2150. Our study compares results generated with three 3-D Earth rheology models; EM1_M and EM2 have a comparable viscosity range, while EM1_L has the lowest viscosity values under the ASE. We find that uncertainties associated with Earth structure are significant and can contribute up to multiple metres of uncertainty in predicted sea level by the end of a 100-year simulation (Fig. 6). Furthermore, additional uncertainty arises from the model of viscoelastic behaviour. We adopt a viscous Maxwell rheology, but studies suggest that incorporating a short-term transient component of deformation may result in even faster viscous deformation (e.g. Pollitz, 2019).

Finally, we note that the required resolution of the GIA model grid will depend on the resolution of the seismic model used to construct the 3-D Earth structure model. Earth structure is currently resolved in seismic tomography models in Antarctica at length scales of $\mathcal{O}(100\ \text{km})$ or greater (Lloyd et al., 2020; Lucas et al., 2020), but as further improvements in the resolution of seismic tomography emerge, variations in Earth properties at even shorter spatial scales may be revealed and need to be represented in GIA models. However, given the smooth nature of viscoelastic deformation and geoid changes, we expect that the wavelength of ice loading variations will remain the determining factor of surface GIA model grid resolution requirements.

**5.3 On GIA model setup**

In choosing a method for representing a finer resolution load grid onto a coarser GIA model grid, we found that it is important to consider how the model itself discretizes the load, and the input load interpolation schemes. Here, it is worth noting that our GIA model grid is a tetrahedral grid (triangular grid on surface), and these findings may not translate perfectly to other model grid compositions. Our GIA model grid consists of a uniform global tetrahedral grid that allows for regional patches of refined resolution (also uniform) but does not permit matching of model grid nodes to the input grid. For our experiments, by comparing the volumes of the input ice calculated on the input and GIA model grids, we found that the in-built Poisson interpolation scheme (Latychev et al., 2005) performed better in interpolating the finest resolution load grid onto the model grid compared to other tested schemes, suggesting that an understanding of the method in which the load in mapped onto the model grid nodes is important. Additionally, we note that considerations such as the resolution of the input ice-sheet model

and treatment of the ice cover outside the region of interest also have an influence on the final GIA model predictions (see Supplementary Section S1) and should be explored further in future studies.

## 6 Conclusion

In this study, we present a comprehensive analysis of the influence of grid resolution on GIA-model predictions in response to ice cover changes in the ASE over the modern satellite era and through the 21$^{st}$ century. We adopt a range of Earth models including models that capture lateral variations in Earth structure based on seismic tomography and GNSS analyses. These experiments showed that: (1) the grid resolution error introduced through adopting a 15 km grid relative to a 1.9 km model grid remains within centimetres to decimetres (which can reach up to 20% of the total signal along the grounding line) throughout our simulations; (2) the grid resolution error is the highest in the vicinity of the grounding line for purely elastic deformation cases, and along past and current grounding lines for viscoelastic Earth models, and is primarily associated with the representation of the surface load; (3) results with grid refinement beyond 3.75 km converged in our simulations, even when adopting a 1 km resolution input load, and this likely represents a conservative lower bound since the next coarser grid we considered was 7.5 km. The errors associated with the choice of grid resolution will decrease with time for longer simulations as the extent and magnitude of ice loss and associated Earth deformation and sea-level change increase. Comparison of simulations adopting elastic and 3-D viscoelastic Earth models demonstrate that the contribution of viscous deformation can be up to tens of metres over the 21$^{st}$ century, or > 60% of the total deformation signal. Furthermore, uncertainties in Earth properties can contribute up to several metres of error. This indicates the importance of considering viscous deformation when modelling GIA over decadal to centennial timescales in the ASE. In comparison, the error due to grid resolution is negligible, especially for grids of spacing of 3.75 km and less.

To supplement these findings with realistic ice loading, we conducted a sensitivity test with cylindrical loads with radii from 16 km to 0.5 km and grid resolutions from 7.5 to 0.5 km. These experiments indicated a minimum 1:3 ratio between the required grid resolution and the load radius (i.e. grid size should be $\leq \frac{1}{3}$ of the load radius) to minimise model grid resolution error. However, no significant spatially isolated loads occur in our adopted observation- and model-based ice loss scenarios, and a preliminary examination of other ice observation and modelled products suggest that significant ice loss with < 5 km wavelength is rare in the ASE. These results, taken together, support the conclusion that km-scale resolution in GIA modelling is generally not necessary. However, as higher resolution sub-km ice observational and dynamic ice model grid products are released, this guidance may need to be revisited.

**Code/Data Availability**

We have made all model output from the sensitivity tests and more realistic simulations available on a public repository at https://osf.io/2vmqh/?view_only=4d08e562720941688e0644b80781eeaa. The 3-D GIA model adopted here has been used in

numerous previous studies, questions regarding the model or requests for additional output can be discussed with the corresponding author and K.L., the developer of the code. Additional data related to this paper may be requested from the authors.

**Author Contribution**

J.X.W.W. and N.G. developed the ideas and experiments in the study with input from K.L. and H.K.H. K.L. contributed to

designing the experimental set up. J.X.W.W. performed the simulations and analysis. J.X.W.W. and N.G. wrote the original text and all authors contributed to revisions.

**Competing interests**

The authors declare they have no competing interests.

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

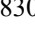

**Figure 1. Grid and Earth model configuration. (a-b) Configuration of the tetrahedral grid in the finite-volume 3-D GIA model with regional refinement, used for observational and modelled ice loading scenarios. (a) shows a cross-sectional view of the regional refinement along ASE. (b) indicates areas of grid refinement across Antarctica with a surface grid resolution of 7.5 km over all**
**Antarctica in black, 3.75 km over a section of West Antarctica in magenta and 1.9 km in the ASE (light blue square). (c-d) Logarithmic viscosity perturbation map at depth 200 km for low upper mantle viscosity model EM1_L over (c) Antarctica and (d) our study region in the Amundsen Sea Embayment. Values are relative to a reference 1-D profile with upper mantle viscosity of 1 x $10^{20}$ Pa s, and lower mantle viscosity of 5 x $10^{21}$ Pa s The black line delimits the edge of the Antarctic ice shelf including the extent of**

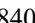

marine-based ice, and the gray line shows the location of the grounding line from Bedmap2 (Fretwell et al., 2013). Transparent patch in (c) contains no data on mantle viscosity as it the region contains lithosphere at 200 km depth.

**Figure 2. Ice loading scenarios and corresponding elastic sea-level predictions in the ASE. (a – c) Total ice thickness change in meters predicted from (a) 1992 to 2017 in the observation-based ICE-SH ice model (Shepherd et al., 2019), and from 2000 to 2100 in the (b) ICE-GOL (Golledge et al., 2019) and ICE-RD (DeConto et al., 2021) ice model projections. (d – f) show the predicted sea-level change**

**in meters with an elastic Earth model associated with the ice cover changes shown in (a-c). (g– i) as in (d-f) but adopting 3-D viscoelastic Earth model EM1_L. All sea-level predictions were performed on a 1.9 km resolution grid. The black and blue line indicates final and initial grounding lines, respectively, for each simulation. Each frame is annotated with the maximum and minimum value within the frame. Note that the colour bars change across each frame.**

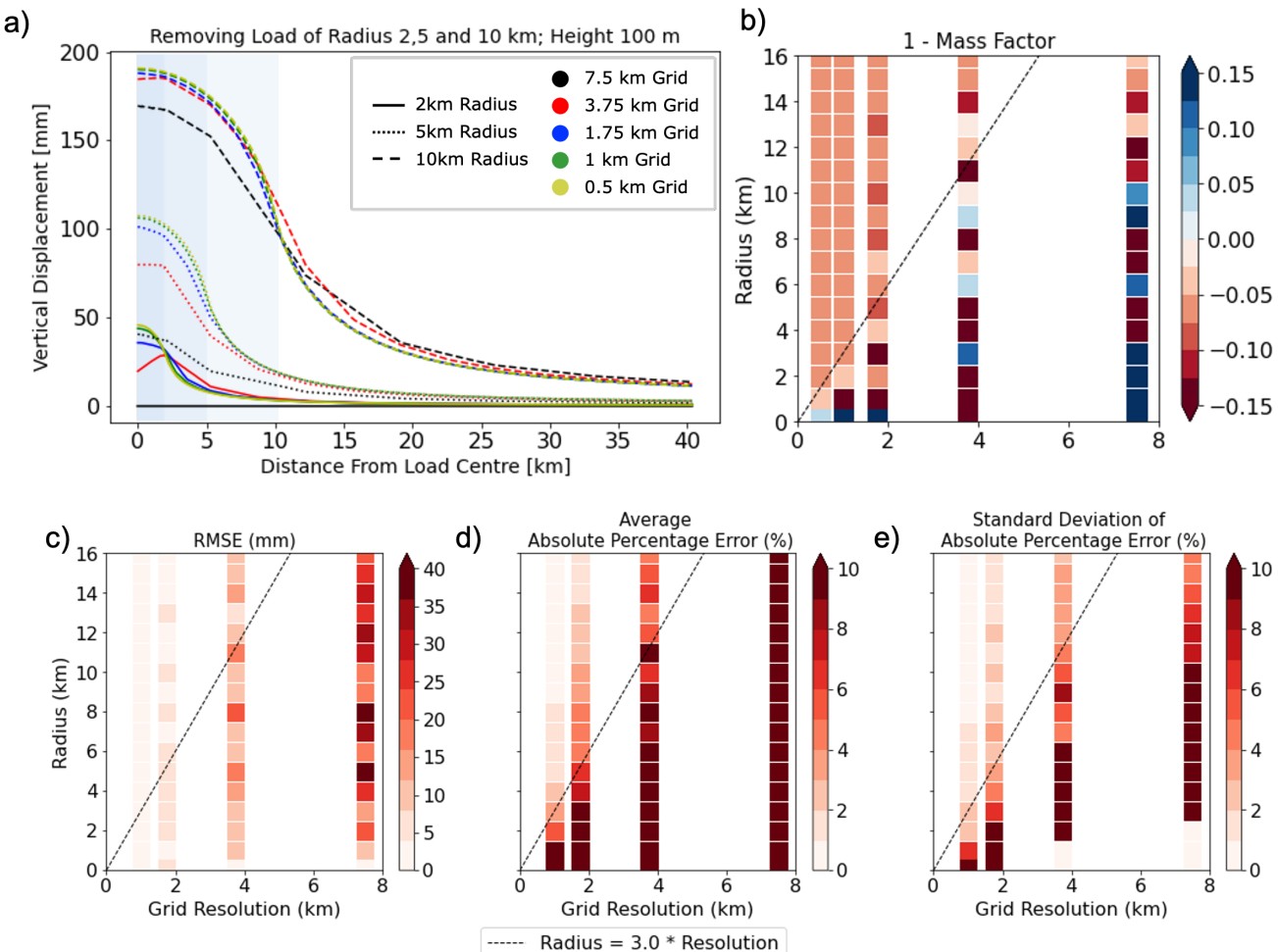

**Figure 3. Idealized sensitivity experiment of the effect of surface grid resolution on GIA model calculations with elastic bedrock deformation due to instantaneous removal of cylindrical ice loads. Cylinders are all of unit height 100 m, and radius from 0.5 to 16 km. Five grid resolutions applied within an area of minimum 40km width were tested: 0.5, 1, 1.75 km, 3.75 and 7.5 km (Figure S1). (a) Transect of bedrock deformation for removal of ice cylinders with unit height and radii of 2 km (solid lines), 5 km (dotted) and 10 km (dashed lines). (b) – € Results of a suite of simulations adopting ranges of ice cylinder radii and grid resolutions. (b) Colours indicate 1 minus the Mass Factor, [ 1 – Mass Factor ], where the Mass Factor is the ratio of the theoretical mass of the load and the mass of the load represented on the given model grid. 0 represents a scenario where the model grid perfectly represents the mass of the idealized load, whilst positive (blue) and negative (red) values indicate the load mass is over- and under-represented by model grid resolution, respectively. (c) Root mean square error across the suite of simulations (mm). (d) Average absolute percentage error (%). (e) Standard deviation of the absolute percentage error (%) of the given test from the finest 0.5 km resolution model result. Errors displayed on (c), (d), (e) are calculated within 2 km of the loaded region. Dashed black lines represent the 1:3 ratio between the surface grid resolution and idealized load cylinder radius whereby average absolute percentage error becomes < 7 ± 3 ($\sigma$) % for all scenarios. In panels (b) – (e) the colour bars are saturated according to the arrows on the respective colour bars.**




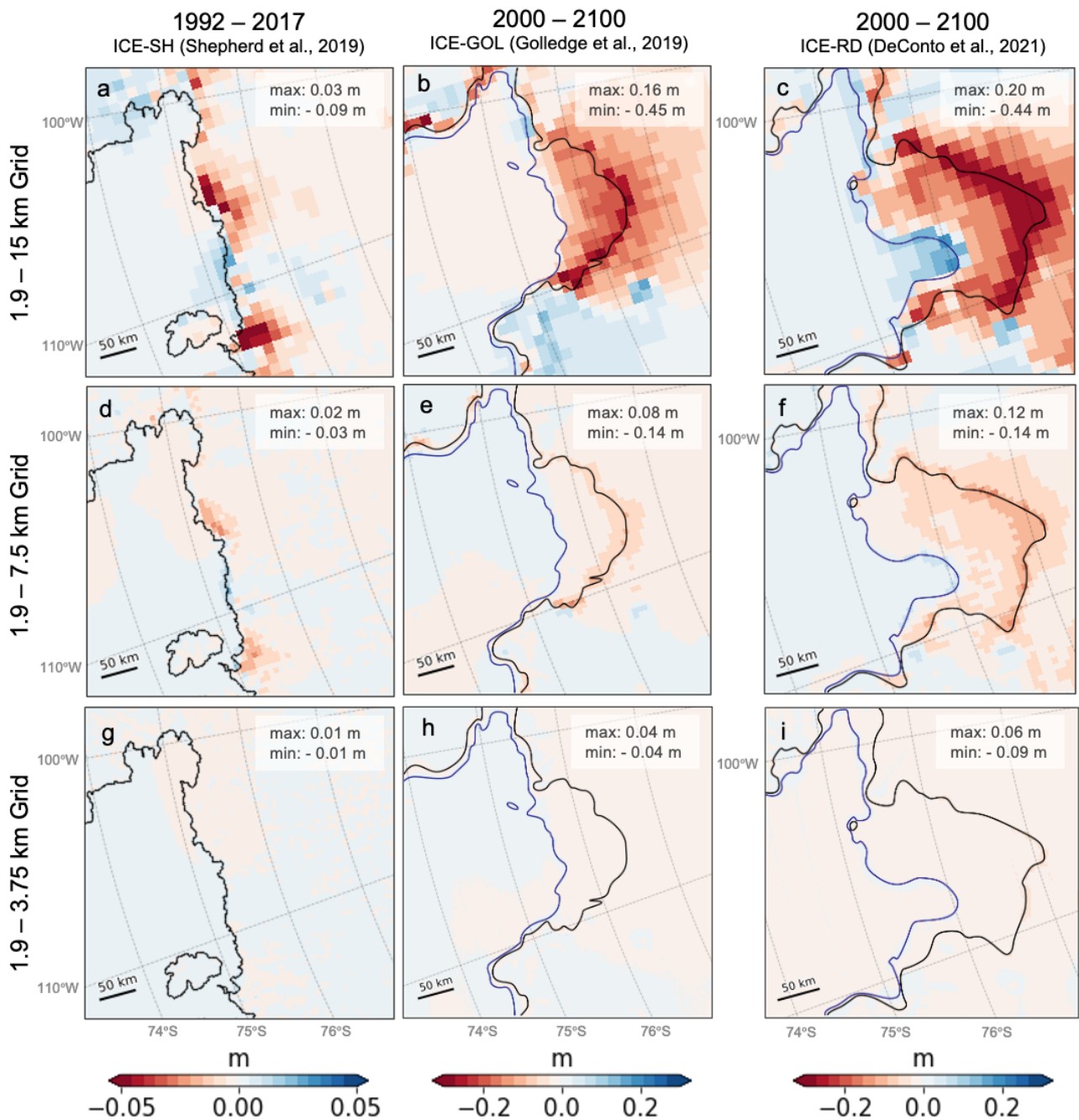

**Figure 4. Influence of grid resolution on elastic sea-level predictions in ASE.** Difference in predicted sea-level change in meters between (a-c) 1.9 and 15 km, (d-f) 1.9 and 7.5 km; and (g-i) 1.9 and 3.75 km resolution GIA model simulations with a purely elastic Earth model across the times indicated at the top of the column for ice loading scenarios (from left – right) ICE-SH ICE-GOL and ICE-RD. For each ice retreat scenario there is a different colour bar since the magnitude of error due to grid resolution differs. In some panels, the colour bar is saturated. The black and blue line indicates final and initial grounding lines, respectively, for each simulation, and each frame is annotated with the maximum and minimum values within the frame.

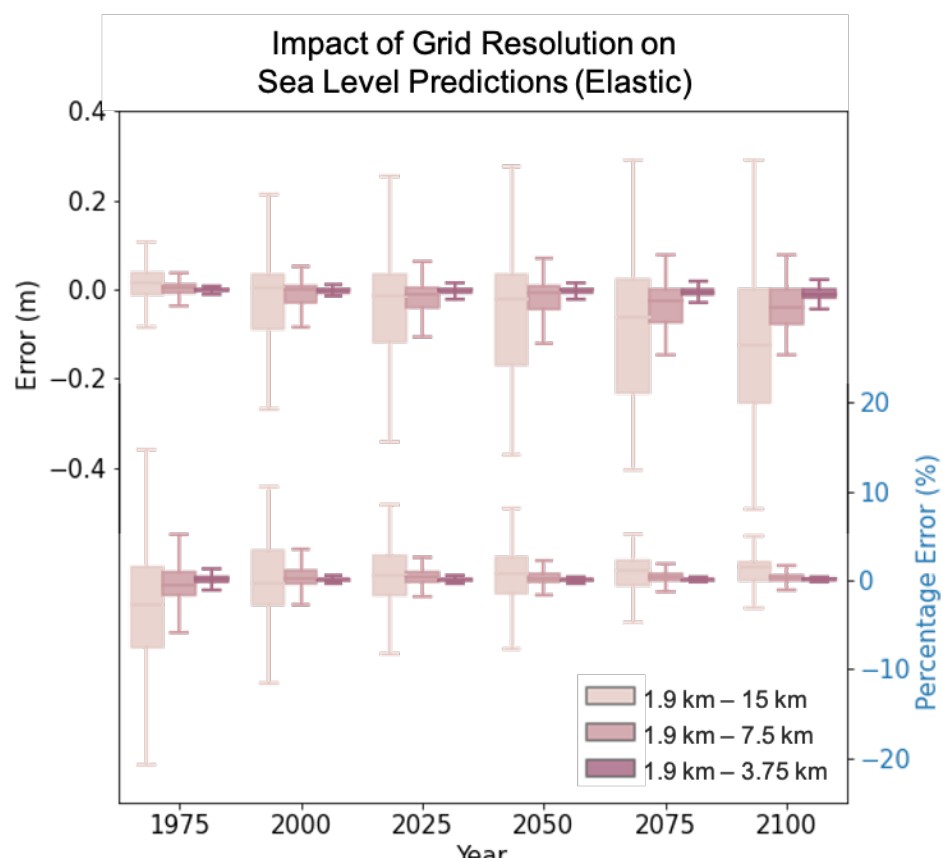

**Figure 5. Evolution of error in elastic sea-level predictions due to grid resolution from 1950 to 2100 with the 1km input resolution ICE-RD ice model. Box-whisker plots of the error and percent error (see methods) calculated from the difference in predicted sea-level changes from the start of the simulation to the indicated time within 10 km of the grounding line at that time between a simulation with 1.9 km resolution and simulations adopting 15 km (light pink), 7.5 km (medium pink) and 3.75 km (dark pink) grid resolutions. The box represents (from bottom to top) the 25th percentile, median and 75th percentile of the distribution, whilst the whiskers represent the "minimum" (25th percentile – 1.5 x the interquartile range) and "maximum (75th percentile + 1.5 x the interquartile range). Error (m) is the difference between sea-level predictions from the highe–r - lower resolution simulation. Percentage Error (%) is calculated as 100* $(SL_{1.9km} - SL_{lowres})/SL_{1.9km}$ for each grid point.**


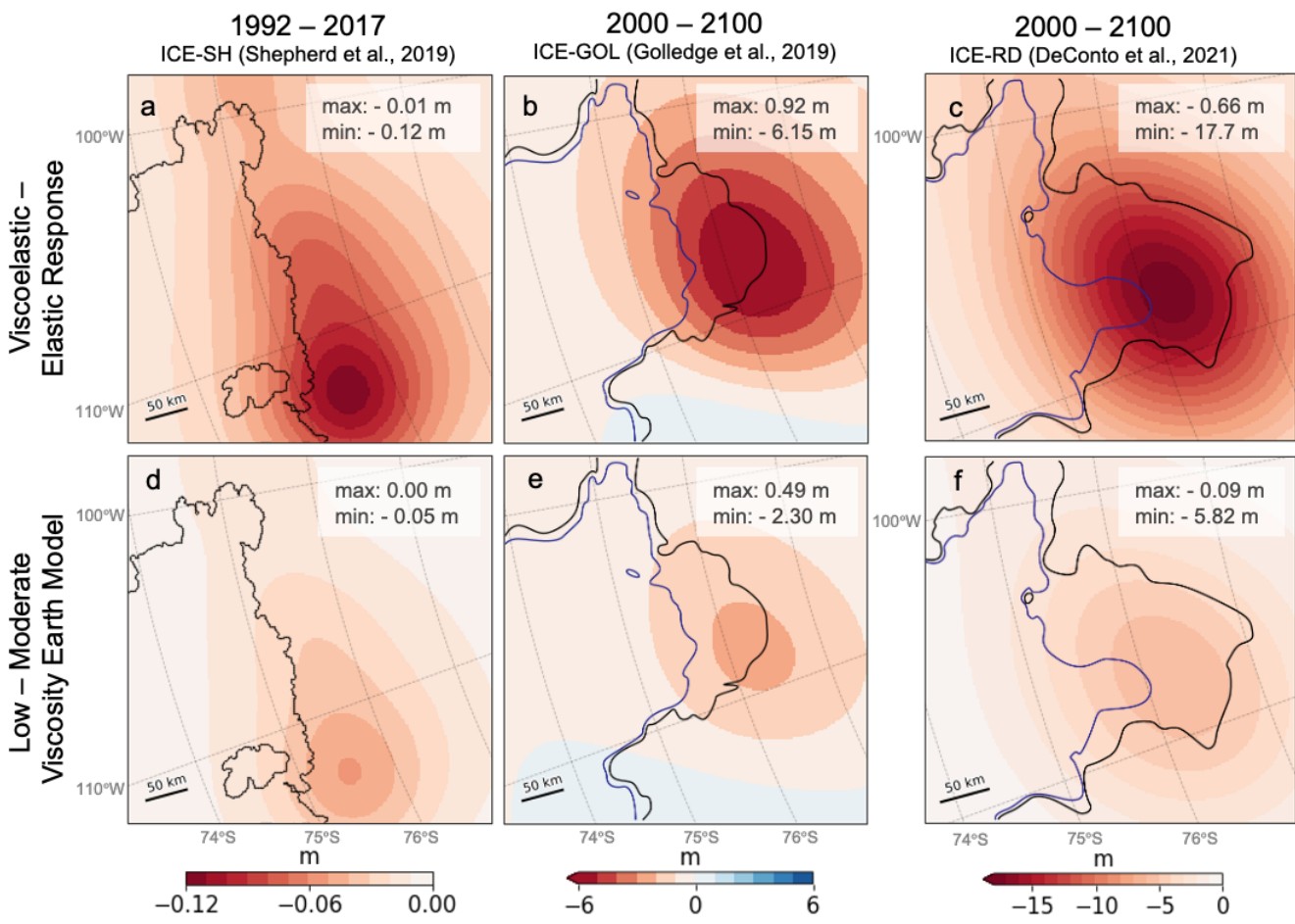

**Figure 6. Influence of incorporating viscous behaviour and uncertainty in viscoelastic Earth structure on sea-level predictions. Frames (a-c) shows the difference in sea-level change predicted from simulations adopting 3-D viscoelastic Earth model EM1_L and an elastic Earth model. Frames (d-f) shows the difference in sea-level change predicted from simulations adopting two different 3-D viscoelastic Earth models EM1_L and EM1_M. Note the difference in scale. Time frames and ice models are as indicated at the top of the columns. In each frame, the black and blue line indicates final and initial grounding lines respectively for each simulation and is annotated with the maximum and minimum data value within the frame. All simulations adopt a 1.9 km grid resolution.**

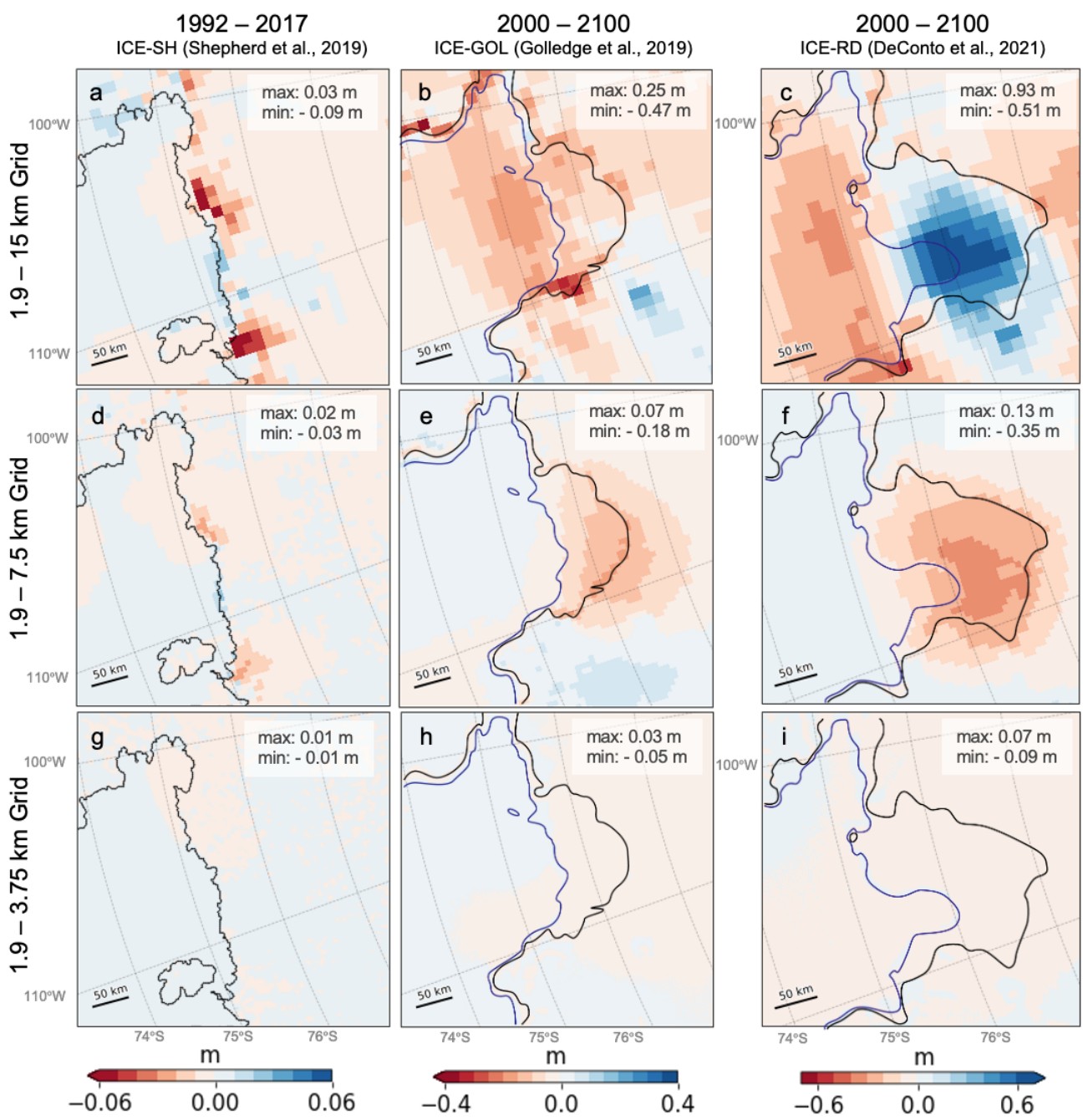

**Figure 7. Influence of grid resolution on viscoelastic sea-level predictions in ASE. As in Figure 4 but adopting 3-D viscoelastic Earth model EM1_L.**


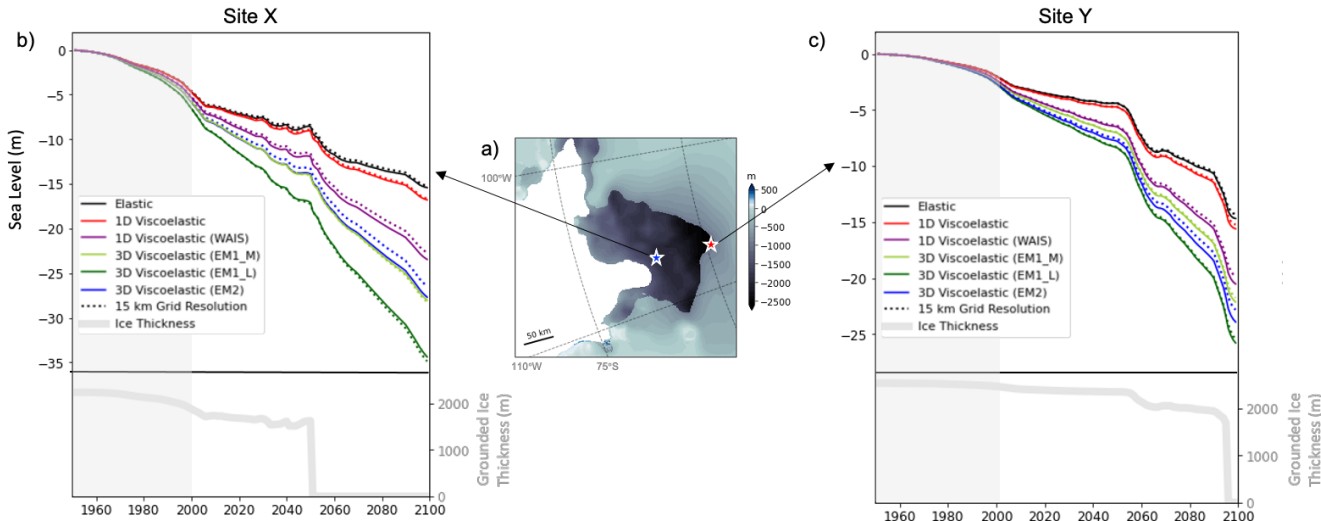

**Figure 8. Evolution of site-specific sea level in simulations adopting a range of model resolutions and Earth models. (a) ice thickness change from 2000 to 2100 predicted in the ICE-RD simulation. This frame is identical to figure 2c. (b) coloured lines show predicted sea-level change, in meters, at Site X that experiences the maximum viscous uplift in the 1.9 km resolution simulation, shown by the**
**blue star in frame (a). Each coloured line represents, respectively,the simulations with a purely elastic solid Earth response (black lines), viscoelastic solid Earth response based on a global average 1-D Earth model (red lines), a 1-D Earth model best-fit for the WAIS from Powell et al. (2021) (purple lines), a low viscosity 3-D Earth model EM1_L (dark green lines), a moderate viscosity 3-D Earth model EM1_M (light green lines) and 3-D viscosity Earth model EM2 (blue lines). Solid lines are for simulations performed at 1.9 km resolution, and dashed lines adopt a 15km resolution. The thick gray line represents the evolution of grounded ice thickness**
**at the respective sites. (c) is as in frame (b) but for Site Y along the final grounding line position at 2100 that experiences the greatest ice thickness change, labeled by the red star in frame (a).**

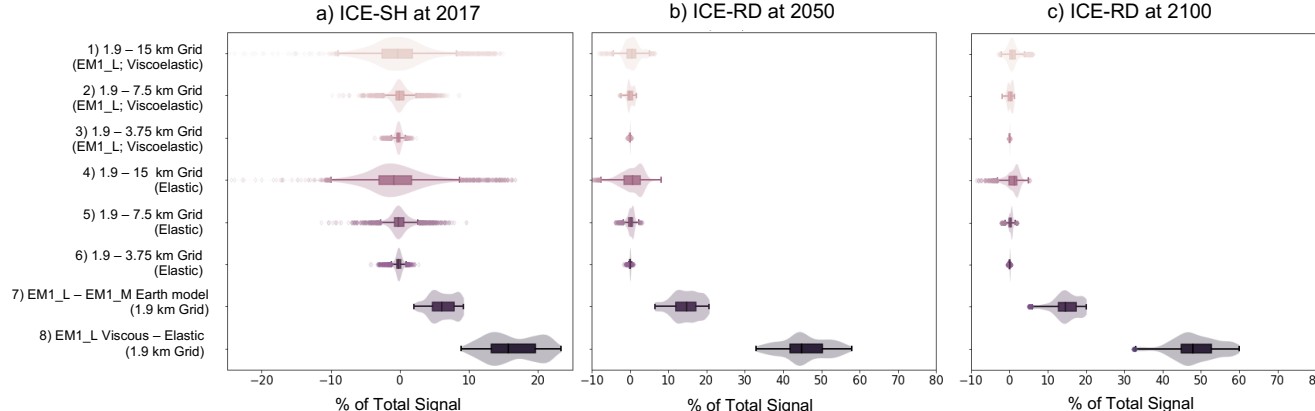


**Figure 9. Comparison of factors contributing to differences in sea-level predictions in this study. Each distribution represents the influence of the specific factor across points within 10 km of the grounding line at the specified year of the simulation. For each factor, the "% Total Signal" is calculated as $(SL_A - SL_B)/SL_B *100\%$, where simulations A and B are described as per the labels on the y-axis with the form "A – B" (e.g. for the distribution described as "1.9 – 15 km Grid", A refers to the 1.9 km grid simulation**

**and B refers to the 15 km grid simulation). Distributions are presented for a) ICE-SH ice model from 1992 to 2017; b) ICE-RD ice model from 1950 to 2050; c) ICE-RD ice model from 1950 to 2100. Eight factors are compared in this figure, as labelled on the y-axis. To visualise the distribution, we plot a classic box-whisker diagram overlain with a density curve. The edges of the box represents (from left to right) the 25th percentile, median and 75th percentile of the distribution, whilst the whiskers represent the "minimum" (25th percentile – 1.5 x the interquartile range) and "maximum" (75th percentile + 1.5 x the interquartile range). The**

**diamonds outside the whiskers represent outliers.**