# Peer review of "Resolving GIA in response to modern and future ice loss at marine grounding lines in West Antarctica"

_The Cryosphere, 2021_

## Author Comment (AC1)

We thank the reviewers for their thoughtful and constructive comments, which helped us to improve our revised manuscript. We provide our response to reviewers' comments in the following section, leaving the reviewers' original comments in black text and our response in blue text.

**Reviewer 1 (Dr. Samuel Kachuck) General Comments**

The manuscript, "Resolving GIA in response to modern and future ice loss at marine grounding lines in West Antarctica" by Wan et al., addresses what computational grid is required to resolve the local sea level changes caused by solid-earth responses to changes in cryospheric loading, and how those compare to uncertainties in the solid-earth mechanisms (e.g., elastic vs. viscoelastic, poorly constrained viscosities). These questions are particularly salient for those engaged in describing and forecasting the sea level effects of changes in the cryosphere, particularly in areas, like West Antarctica, where such solid-earth feedbacks on the future dynamics of the ice are a significant source of uncertainty.

The authors find that the limiting factor for spatial resolution is the representation of the load, rather than the smoother solid-earth response, and provide a useful rule of thumb for determining what resolution is necessary to compute the elastic response of certain loads. They additionally show how uncertainties in the behavior of the mantle (in particular, the possibility of a low viscosity zone underlying parts of West Antarctica) can generate signal differences far exceeding errors caused by resolution, emphasizing how critical it is to include these processes in modeling and to encourage observational constraints of the mantle rheology.

The manuscript is well presented and I have only a handful of rhetorical suggestions to help with cohesion and flow, and one requested additional model run to resolve an apparent inconsistency in the comparison of the 3D rheologies with the 1D average rheology.

Finally, I would like to encourage the authors to release their model code source publicly, in accordance with the journal's data policy, to "guarantee integrity, transparency, reuse, and reproducibility."

Thank you for the summary of our study and findings. In regards to the final sentence, we agree that this is an important issue and appreciate the reviewer raising it. In our Data/Model Accessibility Statement, we note that Konstantin Latychev, the author and maintainer of the 3D model, can be contacted with questions about the code and requests for additional output, and he is also available to work with other groups to run whatever GIA simulation they are interested in. However, since this code has been developed over decades and applied by a range of scientists in numerous previous studies, our study is not the appropriate venue for sharing it publicly for the first time. Note that the coauthors on this study and other users and contributors to the code are actively engaged in community discussions and efforts to make GIA modeling and results increasingly accessible in future. For example, Gomez and colleagues organized an online seminar series over the last year on the theme of ice sheets, sea level and GIA in which GIA model benchmarking and accessibility was a central theme. Discussions on this topic continued at the PALSEA meeting and follow up action will be an ongoing community effort.

**Specific comments**

Comment on the 1D Viscosity: The vertically averaged GIA model you use to compare with the 3D viscosity models is for the whole of Antarctica and therefore not reflective of the viscosity underlying your region of interest (the Amundsen Sea). It makes it difficult for me to evaluate claims like the one made in Line 378 that there is a dependence on the 3D structure within the Amundsen Sea domain or in Line 404 that a 1D viscosity, rather than one with simply a long relaxation time, is closer to the purely elastic model. I appreciate that you quantify the importance of GIA from outside the Amundsen Sea (L313 and Supplemental S1) as being about 6% of the central response, and so might be hesitant to apply a lower viscosity 1D average to the globe. However, if you were to focus only on load changes in the Amundsen Sea, is there not any 1D viscosity profile or relaxation spectrum that, within

other uncertainties, reproduces? This would greatly strengthen the cited arguments and the comparison you make in Figure 8.

We agree with the reviewer on the importance of distinguishing the impact of adding lateral variations in Earth structure from the impact of the lower viscosity upper mantle and thinner lithosphere of the 3D model in the Amundsen Sea compared to the globally averaged 1D model we considered. We note that the question of whether a 1D Earth model can represent the response of the Earth in a zone of complex Earth structure is explored in detail in a study in review by Evelyn Powell et al. titled "The robustness of geodetically-derived 1-D Antarctic viscosity models in the presence of complex 3-D viscoelastic Earth structure". To address this comment, we are performing an additional simulation adopting a 1D viscosity profile from the Powell et al. study that is representative of the low viscosity structure in the West Antarctic, and will add these results to Fig. 8. In the revised manuscript, we will revise the sentences mentioned by the reviewer to clarify this issue and refer to the new results.

Comment on ice dynamics and grounding lines: Though the paper's technical scope is focused on modeling local sea level changes due to solid-earth processes, part of the justification is that these sea level processes will affect the dynamics of the ice's grounding line, and are therefore necessary to model in tandem. I think a good place for this discussion could be surrounding the location of the grounding line under different resolutions and physical assumptions (e.g., elastic or viscoelastic mantle). And yet the final grounding lines in Figure 2, 4, 6, and 7 doesn't appear to change at all from case to case - the variation in the sea level (almost 20 m near the grounding line in Figures 2i,f) doesn't seem to affect where ice is floating. Is that right? In contrast, when we included the ice dynamics, we found a lag in a grounding line position of about 30 km after 100 years for a similar amount of sea level fall at Pine Island (Figure 3b in Kachuck et al., 2020). Considering this will significantly affect the resolution error near the grounding line that you show in Figure 9, and I think is worth discussing.

The goal of this study is to isolate the influence of grid resolution on GIA model predictions, and our experiments are therefore based on a stand alone GIA model with a prescribed ice model input at varying resolutions, rather than a dynamic ice model that responds to the ongoing sea level changes. Hence, we do not incorporate a coupled ice sheet-sea level model run and analysis of how the modelled grounding line is affected by the solid earth response from GIA predictions is outside the scope of this paper.

We further clarify this by adding the following to the methods section of the revised manuscript: "Note that we adopt a stand alone GIA model throughout this study with the purpose to inform the set up of coupled ice sheet - sea level modeling studies, but we do not model the feedback of GIA on ice sheet dynamics explicitly. However, other work (e.g. Kachuck et al., 2020; Gomez et al., 2015) suggests that the scale of differences in GIA simulated at different resolutions and adopting different Earth structure models here will be large enough to alter the timing and magnitude of grounding line migration in a coupled modeling context."

**Line-by-line**

L35: Instead of "consists of... and...", you might indicate that a viscoelastic material's response begins as that of an instantaneous elastic solid, and transitions to a longer-timescale viscous-like relaxation. For heuristic and analytic purposes we sometimes treat them as separable and additive.

**Thank you for this suggestion, we have incorporated it into the text.**

L60: "Viscous effects due to ongoing ice loss ... have not been included in recent high resolution coupled projections": You could cite Kachuck et al (2020), Coulon et al. (2021), or De Conto et al. (2021) as recent examples of coupling ice dynamics to viscous vertical land motion in projections. The

first has high spatial resolution, with 500 m grid spacing near the grounding line, for projections with viscoelastic feedback of Pine Island Glacier in the Amundsen Sea Embayment. The others are Antarctic-wide and determine that the ELRA model, and modifications thereof, are satisfactory for constraining large-scale sea levels.

**Thank you for suggesting these very relevant papers, we have included this in the text to reflect recent developments in high resolution coupled modelling.**

L246: An image of the ice load and topography would really help in understanding this description.

**We will incorporate a figure of this in the supplementary material**

L255: The Scaling Factor used in Figure 3b is not introduced in the main body text and doesn't appear again in the results focused on the Amundsen Sea. I would recommend integrating it more (see next note). Additionally, "scaling factor" is used previously to refer to the construction of viscosity profiles from seismic data (e.g., L179), muddling the usage. Another term might make this clearer.

**We are grateful that the reviewer has pointed out the overlap in usage of the "scaling factor" terminology, we have renamed the scaling factor used in figure 3b to "mass factor". We have also incorporated further discussion of the Fig 3b results in the concluding paragraphs of section 3.1.**

L322: I could use more elaboration here on the differences between representing the ice load and computing the elastic response, and maybe some tie-in with the Idealized Experiments results. Is the idea that we need finer resolution for better representation of the grounding line (i.e., where the ice load disappears), but not necessarily for the **smoother earth response**, which is consistent with your findings in the idealized experiment? If so, could you use the Scaling Factor from Figure 3b to explain this? In those earlier experiments, though, am I right in understanding that you didn't have any ice-ocean interaction? Does the ocean load smooth that boundary?

In response to the latter half of this comment, that is correct, as further emphasized above, the experiments run in this study are stand-alone GIA model simulations with no ice-sheet interaction. But the reviewer is correct in pointing out that high resolution is necessary for better representation of where ice loading is occuring, and less for capturing the smoother response of the solid earth. We appreciate the way the reviewer has articulated the difference and have adjusted the text to adopt this wording.

L353: It is hard for me to visually evaluate this statement across the different scales of the colorbars.

We have adopted a consistent colour bar for ICE-GOL and ICE-SH respectively for the elastic and viscoelastic sea-level pattern Fig. 2. Unfortunately for ICE-RD a consistent colour bar would mean that some of the detail behind the sea level pattern would be lost, so we have decided to keep the different scale for Fig. 2f vs 2i. The difference in magnitude of the sea level fall between 2f and 2i is evident in the large difference in colour bar range.

L369: "A viscous process" is vague. Could you specify that this is a result of assuming viscous incompressibility, or whatever other viscous consequence you are referring to?

Thank you for pointing this out, we have incorporated the suggested clarification.

L423: This would lead me to conclude that high resolution is not important for coupling viscoelastic deformation to grounding line motion. Do you believe that to be so? Are there conditions you can foresee in which resolution would play a larger role?

Yes we believe this would be the case unless there is a very significant and spatially isolated ice loss. Otherwise, where resolution would play a larger role would be when we have achieved GIA model prediction of such high accuracy from incorporating other factors mentioned that the error due to grid resolution becomes comparatively significant. We also want to emphasise that while analysis of the effect of high resolution on ice dynamic models is outside the scope of this paper, we expect resolution to play an important role in ice dynamic models in some contexts based on the existing literature. We discuss this in the introduction.

L465: Although this feedback on grounding line dynamics is not addressed in this study. Yes, we agree that future studies with coupled ice sheet - sea level models can further elaborate on this.

L489: see comment to L322. Clarification there will also clarify here. Thank you, we clarify this point in and adopt the suggested phrasing in the comment to L322 as well.

**Technical corrections** L136: "structure. which" L475: "reflecting" -> "reflect" L481: missing word in "over time our simulations"

We are grateful that the reviewer caught these technical errors.

**Comments on Supplemental material**

L36: The referenced plot is titled "ASE\_10km-ASE\_1km" rather than "ANT\_10km-ASE\_ANT." Thank you for pointing this out, we have edited the text.

Caption to Figure S1. The grounding line appears to be a shade of green rather than the red as described in the text (which will probably be very difficult for individuals with red-green color-blindness).

Thank you for pointing out this error, we have addressed it in the caption.

Figure S2.2a) It looks like the recorded minimum of -3.21 m sea level occurs outside the domain, given the scalebar.

It falls just outside the domain of the plot.

Figure S2.2b) max and min are written as negatives, but look on the scalebar as positive. Thank you for pointing this error out. We have addressed it in the figure.

Citation: https://doi.org/10.5194/tc-2021-232-RC1

**Reviewer 2 (Dr. Grace Nield) Comments:**

**Summary**

"Resolving GIA in response to modern and future ice loss at marine grounding lines in West Antarctica" by Wan et al. In this study the authors seek to find out what GIA model resolution is required to accurately capture viscoelastic deformation in response to present-day and future ice loss in the Amundsen Sea Embayment of West Antarctica. They conduct several experiments to determine this: 1) the elastic response to an idealised cylindrical load; 2) the elastic response to realistic ice loss; 3) viscoelastic response to realistic ice loss with a 1D earth structure and three 3D earth models. The results are given in terms of percentage error for each grid resolution tested and the authors find that errors converge at a resolution of 3.75 km, or three times the radius for the idealised experiment. Furthermore, the results show that the error from neglecting viscoelastic deformation over short time scales, or from adopting different 3D earth models, is far larger than the error from grid resolution. This topic is timely given the efforts within the GIA community to improve models. High-resolution models are very computationally expensive, prohibitively so for most, and quantifying the error for different grid resolutions is very valuable. The paper is well written and clearly organised but would benefit from some small amendments to the text as suggested below.

**General Comments**

1) The main conclusions from this study state that the GIA model grid resolution needed for an isolated cylindrical load is three times its radius, and that for realistic ice loading in the ASE a grid resolution of 3.75 km is required to achieve an error of 2%. I think the paper could be improved by adding a qualitative discussion of how these findings might apply in other areas. Line 344 states "for most applications, errors of less than 5% can be achieved with a 7.5 km grid, and errors of less than 2% with a 3.75 km grid". It would be beneficial to have a short section in the discussion expanding on this, discussing whether this rule of thumb is limited to ASE, or marine grounding lines, whether it might apply in other areas of Antarctica where present-day ice change is occurring, or how different spatial scale of ice loading change might affect this general rule.

Thank you for the insightful suggestion. To address this, we added to the Discussion section that "Whilst we focus on the ASE where there is both ongoing and expected future ice loading and complex Earth structure, since the resolution error is primarily associated with the representation of the ice load, the general conclusions on the resolution required and results of the sensitivity experiments can be applied to any area of active, localised ice loading, for example in other parts of Antarctica, in Greenland or in the vicinity of smaller glaciers."

2) In this study the authors find that representation of the ice load on the GIA model grid leads to higher error than the grid resolution itself, however, there is no discussion of how representation of the Earth structure within the model grid might impact the results. Particularly in Section 2.2 and 2.3, what is the resolution of the seismic tomography data that is used? How does this compare with the vertical resolution of the grid with depth, how much is it being down sampled? Perhaps the results from the tests mentioned on line 158/159 could be included in the supplementary information.

This is an important consideration. We have added the following text to the methods section to provide further detail on the resolution of the Earth models used: "ANT-20 is provided on a 25 km grid at the surface, with distance between the points decreasing with depth, and provided at depth slices in 5 km intervals from 0 to 800 km depth. S362ANI is a 3-D global anisotropic shear wave velocity model for the whole mantle, extending from 25 km depth to the core-mantle-boundary defined at 2891 km depth, provided at a 2° lateral grid resolution at the surface. We note, however, that the spacing of the seismic model grid is distinct from the scale of the Earth structure variations captured by the model. While quantitatively assessing the latter remains an outstanding challenge in seismic tomography, we expect the resolution of the ANT-20 model to be O( > 100 km) in the upper mantle and coarser at greater depths, and thus well represented by the GIA model grid."

We have also added a discussion of this issue in section 5.2: *"Finally, we note that the required resolution of the GIA model grid will depend on the resolution of the seismic model used to construct*

the 3D Earth structure model. Earth structure is currently resolved in seismic tomography models in Antarctica at length scales of O(100 km) or greater (Lloyd et al., 2020; Lucas et al., 2020), but as further improvements in the resolution of seismic tomography emerge, variations in earth properties at even shorter spatial scales may be revealed and need to be represented in GIA models. However, given the smooth nature of the GIA response, we expect that the wavelength of ice loading variations will remain the determining factor of surface GIA model grid resolution requirements."

**Specific Comments**

Line 21: The authors repeatedly refer to a "spatially isolated load". The use of this term is particularly important since once of the main conclusions of the paper – a grid resolution needed of 3 times the radius of the load – only holds for cylindrical loads. For clarity, consider changing this term to "spatially isolated disc/cylindrical load".

Thank you for bringing this point up - In discussion of the rule of thumb we pointed out again that we consider only cylindrical loads as follows: *"The results from this analysis of spatially isolated cylindrical loads provide a rough estimate of the magnitude of error one can expect from a given model resolution and loading scenario, and can serve as a guide for selecting the appropriate grid resolution for a given problem."*

Line 57: please quantify short spatial scales, e.g. 10s of km, or 100s of km. We have quantified short spatial scales as less than 100s of km.

Figure 1c: is the white area saturated? Please add to colour bar

The white area is not an area of saturation. Instead, it is a region with no data as the thick lithosphere in East Antarctica extends down to > 200 km depths. We have clarified this in the figure caption.

Figure 1d: scale bar in the figure is not clear. The choice of colour bar makes it difficult to see the regional variation in viscosity.

Recognising these constraints, we chose to retain this colour bar to remain consistent with Fig. 1c. Each discrete colour represents  $\sim 0.2 \log(n2d/n1d)$  difference.

Line 151: 70 radial layers – Are these radial layers where properties are assigned, or is this the vertical resolution of the grid? On line 153 it says layer boundaries at 12, 25, and 43km, again are these the boundaries of material properties or grid resolution?

We designed the grid to ensure there are radial layers on every layer where material properties are assigned in the 1D reference model STW105. We have clarified this further in the text, stating "*The radial layersof the grid are defined to respect the unconformities in material properties of the radially varying (1-D) seismic reference model STW105 (Kustowski et al., 2008), with the shallowest layers at 12, 25 and 43 km depth".*

Line 157: 29 million nodes is a lot! It would be interesting to add here what the run time of this model is.

Our simulations take ~ 65 CPU hours (30 minutes on 134 nodes) to run per time step on a high performance computing cluster. We have added this information to the text here.

Figure 3a: consider adding solid/dotted/dashed lines to the key for the 2/5/10km ice, or perhaps label. Figure 3b: x-axis label - Grid Resolution km, not m?

Figure 3b-e: y-axis would look better labelled at 2km intervals rather than 2.5 km intervals. Thank you for the great suggestion and for pointing out the error in the figures. We have made the suggested changes for Fig. 3b Line 260: "grid points" do you mean nodes? Is the load applied at the nodes or the centre of the elements?

Yes, we are referring to grid nodes and we have clarified this point in the text. The centre of the cylindrical load on an arbitrary model grid node in the ASE (76°S 150°W).

Line 292: "serve as a guide.... appropriate grid resolution for a given problem" I think its important to include here that this guide is restricted to isolated cylindrical loads.

As mentioned above, we have highlighted further here in the text that these insights are based on experiments with isolated cylindrical loads.

Line 335-344: There is reference to Figure 5(a) and 5(b) in this paragraph but there are no corresponding panels in figure 5. Fig 5 caption also only states (a). Thank you for pointing this out. We had a Fig. 5b in a previous iteration and neglected to edit it out in the text, this has been addressed.

Figure 6: (and 2/4/7) it would be useful to include the acronyms ICE\_SH etc referred to in the text in the column headings of these figures.

Thank you for this suggestion, we've incorporated it to increase the ease of interpreting our figures 2,4,6,7.

Line 397: This reference is missing from the list.

Figure 8: panels b and c need y-axis labels; the grey colour showing ice thickness in the lower panels is washed out and hard to see. In the caption for panel a) this is not identical to figure 3c, perhaps figure 2 instead.

Thank you for spotting these mistakes, we've addressed these in Fig. 8.

Line 402: It is confusing to label these two sites A/B since the panels in the figure are labelled b and c. Perhaps change to 1/2 or X/Y, since I think this is the only time they are referred to anyway. Thank you for the suggestion, we have incorporated it in the text.

Line 403: it is not clear what the authors mean by "spin up time".

This is further clarified with added reference in the revised manuscript: "... represents a time of hindcast spin up to the year 2000 in the ice sheet model simulation (see DeConto et al., 2021) rather than..."

Figure 9: the hollow diamonds don't show up very well on this figure.

Thank you for pointing this out, unfortunately due to the density of the points there is inevitable overlap which makes the hollow diamonds hard to discern. They only occur outside the "whiskers" of the plot.

Line 560: add "in the ASE" to the first sentence to clarify the region the conclusions apply to. We've addressed this in the text.

Line 564: would be useful to quote the percentage error in conclusion (1) We've addressed this in the text.

The following paper is relevant and worth citing:

Blank, B., Barletta, V., Hu, H., Pappa, F., & van der Wal, W. (2021). Effect of lateral and stress-dependent viscosity variations on GIA induced uplift rates in the Amundsen Sea Embayment. Geochemistry, Geophysics, Geosystems. https://doi.org/10.1029/2021GC009807

**Technical Comments**

Line 158: missing word "limited to the surface (and?) a few layers down to 10km" Line 211: incorrect spelling of adopted Line 475: reflect rather than reflecting We are grateful that the reviewer caught these technical errors.

Citation: https://doi.org/10.5194/tc-2021-232-RC2

---

## Editor Decision (ED1)

I would like to thank both reviewers for their constructive comments on this manuscript and also the authors for submitting their responses to the reviewers' comments.

The manuscript investigates the challenges of accurately and efficiently modelling the solid Earth response to mass change of a marine-grounded ice sheet located in a region of complex Earth rheology. This is a well-written paper and both reviews are positive – they highlight the importance of the study and the clear communication of the findings.

The authors have responded to all the points raised by the reviewers, and they note that they plan to carry out an additional simulation. I support this decision – do let me know if you need additional time to complete the simulation and any resulting edits to the manuscript.

With regard to reviewer 1's encouragement that the authors release the model code, the authors provide a well-argued response to this suggestion which is reflected by their text in the 'Code/Data Availability' section. They also note the ongoing efforts within the GIA modelling community (including the efforts of the authors) to increase the accessibility of such codes – thank you for pursuing this and I encourage you to continue working towards this important goal.

In addition to the points raised by the reviewers, I have a couple of small queries:

- Line 316: you mention that you repeat the calculations of fig. 2d-f using a range of grid resolutions, but I found it difficult to identify the resolution used to produce the original results shown in figure 2

- Line 425: "where the *edges of the* boxes represent" (also check figure captions)

- Line 493: "past the grounding line" – specify whether this refers to locations upstream or downstream of the grounding line. In this paragraph, it may also be useful to mention that deformation offshore of the grounding line can be important for ice sheet stabilization if it causes regrounding of an ice shelf

My decision following the initial round of reviews is to publish this article subject to revisions (the process may involve further review by the original reviewers). I invite the authors to submit a revised version of the manuscript that addresses the points raised during the review process.

Kind regards,

Pippa Whitehouse

---

## Author Response (AR2)

We would like to thank the editor for these helpful comments, we sincerely appreciate the thoughtful feedback and have rigorously reviewed the paper to address it. See our responses to the latest comments in blue. Do also note that all the line references mentioned refer to that in the track changes version.

**Editor comments on "Resolving GIA in response to modern and future ice loss at marine grounding lines in West Antarctica" by Wan et al**

I would like to thank the authors for providing a clear response to the reviewers, and for submitting a revised version of their article following completion of the additional model run. I am satisfied that the edits address all the points raised by the reviewers and there is no need to request further review by the original reviewers. However, in reading through the manuscript I have identified a number of issues that require clarification and a couple of areas where terminology is a little inconsistent. These are listed below (line numbers refer to version 3 of the manuscript).

This is an important study that is very thorough in its investigation of the impact of grid resolution and earth model choice when modelling GIA in Antarctica. There is a significant amount of work presented in the article and the conclusions are robustly supported by the results.

My decision is to publish this article subject to minor revisions (review by the editor). Although there are quite a few points listed below, the majority can be resolved very easily. Where I have requested additional detail on the modelling, you are welcome to address this by editing the supplementary material, if suitable.

Thank you for submitting your article to The Cryosphere. Pippa Whitehouse (editor)
* * *
Main points

**Definition of GIA**: The opening paragraph of the Introduction is very useful for introducing the range of processes associated with GIA but there is some discrepancy in how you use this term throughout the rest of text. For example, I think the sensitivity experiments (section 3) just consider solid Earth deformation but in section 4 you present results for sea-level change, which reflect deformation of both the solid Earth and the geoid. In both cases, the results are generally described as representing 'GIA', with no clarification of which components are included. You also use the term GIA to refer to the elastic response to surface load change and the rapid viscous response that is triggered in low mantle viscosity regions such as the Amundsen Sea Embayment – these are not standard definitions, and many people still consider the 'GIA signal' to be the decaying response to long-complete ice mass change immediately following the Last Glacial Maximum. Make sure you are explicit about what processes are captured by your use of the term GIA and, if necessary, clarify which components are included in the different sections of the paper.

We appreciate this point and found it a challenge to decide on terminology that would be interpreted consistently across research areas. We have further clarified or changed the terminology around GIA, sea level change and solid Earth deformation in a number of places throughout the manuscript, notably in the abstract, Introduction, and an added note in Methods Section 2.1 (see track changes).

Note that for the sensitivity tests in section 3, while we plotted the error in elastic deformation for comparison with recent literature, we performed a full sea level calculation with the GIA model described in section 2.1, including changes to the position of the geoid and solid surface. We have

further clarified which components of the sea level calculation are considered in each section of analysis in the following locations:

- Line 296: "Figure 3 summarises the error in the predicted elastic…"
- Line 300: "errors in the solid Earth deformation prediction reported relative to the result…"
- Line 350: "Throughout section 4, grid resolution error is defined as departures of predicted sea level changes from the finest resolution 1.9 km grid resolution result."
- Line 673: "These experiments indicate a minimum 1:3 ratio between the required grid resolution and the load radius (i.e. grid size should be $\leq \frac{1}{3}$ of the load radius) to minimise grid resolution error in predictions of solid Earth deformation."

**Grid resolution vs load size**: the abstract relates grid resolution to load radius, but care is needed when discussing this result because an *increase* in resolution equates to a *decrease* in grid size. It is stated in the abstract that a ratio of 1:3 is required to accurately capture the elastic response of the Earth – it would be useful if you could clarify whether this is an upper or lower bound (noting the care needed when talking about 'resolution'). Also, in some cases you quote a 1:3 ratio and in others a 3:1 ratio (with wording appropriately altered) – please be consistent in how this result is reported.

Thank you for spotting this – we have edited Line 331-333 to report a 1:3 ratio consistently.

We have also clarified whether the 1:3 ratio is an upper or lower bound in the following locations:

- Line 23 (abstract): "we find that a grid resolution at ~1/3 of the radius of the load or higher is required to accurately capture the elastic response of the Earth."
- Line 673: "These experiments indicate a minimum 1:3 ratio between the required grid resolution and the load radius (i.e. grid size should be $\leq \frac{1}{3}$ of the load radius) to minimise grid resolution error in predictions of solid Earth deformation."

**Quantification of error**: in all cases, you report errors relative to the results obtained using the finest- resolution grid (rather than relative to, e.g., an analytical solution). This is clear in most places, but please check for instances where it is ambiguous, e.g. line 24 of the abstract.

This point is clarified in the following:

- Line 25 (abstract): "predicted deformation and associated sea-level change along the grounding line converge to within 5% with grid resolutions of 7.5km or higher, and to within 2% for grid resolutions of 3.75 and higher, …"
- Line 300: "errors in the solid Earth deformation prediction, reported relative to the result…"
- Line 578: "In our simulations, a 3.75 km grid was sufficient to bring errors relative to the finest resolution simulation to < 2% along the grounding line for all scenarios (Fig. 9)."

**Representation of loading**: (i) Figures 2a-c show total ice thickness change, but this does not reflect net surface mass change in locations where there is a transition from marine-grounded ice to ocean because much of the ice load will be replaced by water load. Have you plotted the net surface mass change, and would it be useful to include such plots to aid interpretation of the results? (ii) Errors peak along the location of the final grounding line in experiments ICE-GOL and ICE-RD (fig. 4). This is described as the 'load edge' at various points in the text but if I have correctly interpreted how net surface mass change is computed the grounding line will not necessarily align with the 'edge' of the load. Did you consider whether the misfit along the final grounding line position may be fundamentally related to representation of this transition within the GIA model? (iii) To understand

how you calculate the 'Mass Factor' (fig. 3) a little more information is needed on how the model applies the load. Using the sensitivity experiments as an example; is the same load (i.e. 100m ice) applied to all elements that lie within, or partially within, the footprint of the load, or is the load scaled according to how much of each element is covered by the footprint of the cylinder?

(i)     These plots are meant to represent the ice loads inputted into the model, while the net surface load is an output of the model.  In other words, we need to know how the solid Earth and geoid will adjust after the ice load is removed to understand how the area will inundate with water. This is a non-trivial result to produce without re-running the simulations, and the solution will be different for each different adopted Earth model and grid resolution.  While this would be an interesting calculation to consider, we are weary of the length and detail of our study as is, and feel that this calculation would require substantial added text to properly explain and incorporate the results into the discussion in a meaningful way. We have therefore chosen not to include it.

(ii)    We agree that the grounding line and load edge are not synonymous in the realistic ice loss scenarios and have adjusted the wording accordingly.

(iii)   We have clarified the method for which the model applies the load in Lines 270-274  in the methods section 2.4 as the following: "When inputting a given ice load into the 3-D GIA model, the load mapper algorithm interpolates via a non-linear scheme, the equivalent load acting on each triangular area in the computational grid. Subsequently, an equivalent of 1/3 of the share of the load falling on each triangle grid area incident on the node is summed onto the loaded computational grid node. Within the computational grid triangle area, the load is assumed to be a linear function in triangular coordinates."

Minor clarifications

Place names: rather than 'the West Antarctica' we tend to refer to 'West Antarctica' or 'the West Antarctic Ice Sheet' (similarly for East Antarctica), however, we do refer to 'the Antarctic Peninsula'. 'Central Antarctica' (line 218) is not standard terminology.

We adopted the 'central Antarctica'  phrasing from Heeszel et al. (2016) but agree that it is not standard terminology.  We have revised the place names accordingly.

Section 2.2: I think the 15 km grid is used to produce results for section 4 but not section 3; it would be useful to clarify this somewhere in this section.

Thank you, we have clarified this point in lines 171-174 of the paper.

Line 162: 'incrementally smaller regions' – please clarify whether you use a series of nested regions, i.e. the 3.75km grid is always located within a slightly larger 7.5km grid, which is located within the 15km grid, or whether each higher-resolution grid is inserted directly into the 15km grid.

We have clarified that it is a "incrementally smaller series of nested 3-D regions" in line 173.

Line 164: 'a few layers down to 10 km' – line 158 states that the shallowest layer in the grid is at 12km; do you add extra layers when creating the higher-resolution grids?

Yes, extra layers are added in creation of higher-resolution grids due to the nature of the refinement which is to bisect the grid nodes throughout the 3-D grid which are found over a specific 3D region. We've clarified this point on line 164.

Line 170: is the elastic/density structure used in the purely elastic model the same as that used in the 3D models (described on line 174)? What is the lithosphere thickness used in the elastic model?

We have clarified this in line 185: "For the idealized sensitivity tests in section 3, we adopt a purely elastic Earth model with a 1-D elastic and density structure based on Preliminary Reference Earth Model (Dziewonski and Anderson, 1981)." Note that we discuss the lithospheric thicknesses for the viscoelastic models already.

Line 203: 'close to the preferred value in Kaufmann et al.' – I couldn't work out what this refers to since there is no mention of a scaling factor in the cited article. In general, it is not clear how the scaling factor is applied, making it difficult to derive useful insight from the values quoted here.

We have revised the text in section 2.3 to provide more information on how to relate the scaling factor in our model to the results of Kaufmann et al. (2005), and we added a supplementary section that elaborates further for readers who are interested in the details of this.

Lines 209-210: suggest '...viscosity estimates derived from GPS bedrock uplift rates in three regions' (note that the Antarctic Peninsula is often defined to be a separate region to the WAIS)

Thank you for spotting this, we've made the suggested change.

Line 227: is initial bedrock topography (as well as ice thickness) derived from Bedmap2?

Yes, we've clarified this on line 251.

Section 3.1/figs 3, 4, 7: when calculating the RMSE/percentage error in fig. 3, and the misfits shown in figs 4 and 7, how do you quantify the difference between results determined using different grids, i.e. in situations where output is produced at different resolutions?

For section 3, all results are interpolated onto a 200 m resolution grid over the study region. This is now clarified in line 293. Similarly fr section 4, all results are projected onto the same lat-lon grid – this is clarified in line 179 of methods section 2.2.

Line 320: I think the ICE-SH model covers 25 years, between 1992-2017 (also check caption to fig. 2).

Thank you for spotting this error – we have made the necessary revisions in Line 358 and caption to fig. 2.

Line 373: should include references to fig. 2, not fig. 3?

Thank you for spotting this error – we have made the necessary revisions.

Line 400-401: 'Along the final grounding line...' – based on fig. 7c, I don't think this statement holds.

Agreed, we have removed this line.

Line 403: you mention five earth models, but only four are described in this paragraph. I suspect the fifth model is the 1D/WAIS model described in the next section, but this is not clear.

Yes, apologies for the oversight we have clarified this in line 445.

Line 407: I think the EM1_L and EM1_M models are derived using different 1D viscosity profiles as well as different viscosity scaling factors – this may be worth mentioning.

Yes, this is clarified in line 449.

Lines 437-444: statements about time in this paragraph (e.g. 'after 50 years', 'within decades') are ambiguous because it is not clear whether they are referenced to the start of the model run (in 1950) or the start of the 21$^{st}$ century (as suggested by the opening sentence).

It is from the start of the 21$^{st}$ century at year 2000. This is clarified in line 481.

Line 452: is the 'maximum' value defined by the 75$^{th}$ percentile *plus* 1.5 times the interquartile range (I think 'minus' is correct for the 'minimum' value)? Comment also applies to various figure captions.

Thank you for spotting this error, we have clarified this on line 498-499 and the captions for fig. 5 and 9.

Line 469: 'the difference in predictions associated with earth model configuration lies between ~2-10% ...' – this statement is based on a comparison between results derived using the EM1_L and EM1_M models, which are derived from the same underlying seismic model, i.e., this is not the most extreme comparison that could be made. This statement may require a caveat.

We have clarified this in line 515 "Over the 25-year modern observed ice loss scenario, the difference in predictions associated with earth model configuration between results using EM1_L and EM1_M lies between ~2-10% within 10-km of the grounding line"

Line 482: 'the elastic signal' -> 'the total signal'?

We have clarified this point on line 529.

Line 539: a value of 7 +/- 3 is quoted elsewhere (e.g. line 300, caption to figure 3) – which is correct?

Thank you for catching this, 7+/-3 is the correct value. We have revised this in line 590.

Figures: please add latitude and longitude labels to relevant figures.

We have added the latitude and longitude labels to all relevant figures.

Figures 1c,d and S1: I think the black line marks the edge of the ice shelf and the grey line marks the grounding line – not the 0 m contour, which is typically much further inland.

Yes, we apologise for the error and have clarified the captions accordingly.

Figure 3: (i) 'calculated within 2 km of the loaded region' – does this statement apply to multiple plots in the figure? (ii) The ratio between cylinder radius and grid resolution is quoted as 3:1, not 1:3, in the main text (lines 298-299). (iii) Do the colours saturate in plots b-e?

We have edited the figure 3 caption to clarify the points raised: "Errors displayed on (c), (d), (e) are calculated within 2 km of the loaded region. Dashed black lines represent the 1:3 ratio between the surface grid resolution and idealized load cylinder radius whereby average absolute percentage error becomes < 7 $\pm$ 3 ($\sigma$) % for all scenarios. In panels (b) – (e) the colour bars are saturated according to the arrows on the respective colour bars."

Figure 5: (i) the reference to plot (a) is not needed. (ii) The reference to 'left to right' when describing what is represented by the edges of the boxes does not make sense – plot rotated?

Thank you for spotting this error, we have removed the reference to plot (a), and changed the reference from "left to right" -> "bottom to top".

Figure 8: (i) in the lower panel of fig. 8b, why does ice thickness not reach zero by 2100? My understanding is that the grounding line retreats upstream of this point during the model run. (ii) I suggest splitting the sentence that describes what is shown in plot (b) – it is very long! (iii) Please refer to Powell (2021) rather than the article in review.

The ice thickness includes when the ice becomes floating ice. We have plotted the grounded ice thickness in Fig. 8 instead.

Figure 9: please clarify which earth models were used to produce each set (row) of results.

Good point, we have updated figure 9 axes labels to clarify which earth model was used

Figure S1: please include a label indicating the depth associated with each plot in panels (a) and (b).

Great idea, we have included the depth labels for Fig. S1.

There are occasional small words missing or grammatical errors – please carry out a final check for such issues before resubmitting the article. Please also check that all values quoted in the text agree with values listed in the figures (e.g. line 344).

We have carried out a final check for these issues and made a number of minor adjustments (hyphens, missing words, long sentences, etc.)

---

## Editor Decision (ED2)

**Editor comments on "Resolving GIA in response to modern and future ice loss at marine grounding lines in West Antarctica" by Wan et al**

I would like to thank the authors for providing a clear response to the reviewers, and for submitting a revised version of their article following completion of the additional model run. I am satisfied that the edits address all the points raised by the reviewers and there is no need to request further review by the original reviewers. However, in reading through the manuscript I have identified a number of issues that require clarification and a couple of areas where terminology is a little inconsistent. These are listed below (line numbers refer to version 3 of the manuscript).

This is an important study that is very thorough in its investigation of the impact of grid resolution and earth model choice when modelling GIA in Antarctica. There is a significant amount of work presented in the article and the conclusions are robustly supported by the results.

My decision is to publish this article subject to minor revisions (review by the editor). Although there are quite a few points listed below, the majority can be resolved very easily. Where I have requested additional detail on the modelling, you are welcome to address this by editing the supplementary material, if suitable.

Thank you for submitting your article to The Cryosphere.

Pippa Whitehouse (editor)
* * *
Main points

**Definition of GIA**: The opening paragraph of the Introduction is very useful for introducing the range of processes associated with GIA but there is some discrepancy in how you use this term throughout the rest of text. For example, I think the sensitivity experiments (section 3) just consider solid Earth deformation but in section 4 you present results for sea-level change, which reflect deformation of both the solid Earth and the geoid. In both cases, the results are generally described as representing 'GIA', with no clarification of which components are included. You also use the term GIA to refer to the elastic response to surface load change and the rapid viscous response that is triggered in low mantle viscosity regions such as the Amundsen Sea Embayment – these are not standard definitions, and many people still consider the 'GIA signal' to be the decaying response to long-complete ice mass change immediately following the Last Glacial Maximum. Make sure you are explicit about what processes are captured by your use of the term GIA and, if necessary, clarify which components are included in the different sections of the paper.

**Grid resolution vs load size**: the abstract relates grid resolution to load radius, but care is needed when discussing this result because an *increase* in resolution equates to a *decrease* in grid size. It is stated in the abstract that a ratio of 1:3 is required to accurately capture the elastic response of the Earth – it would be useful if you could clarify whether this is an upper or lower bound (noting the care needed when talking about 'resolution'). Also, in some cases you quote a 1:3 ratio and in others a 3:1 ratio (with wording appropriately altered) – please be consistent in how this result is reported.

**Quantification of error**: in all cases, you report errors relative to the results obtained using the finest-resolution grid (rather than relative to, e.g., an analytical solution). This is clear in most places, but please check for instances where it is ambiguous, e.g. line 24 of the abstract.

**Representation of loading**: (i) Figures 2a-c show total ice thickness change, but this does not reflect net surface mass change in locations where there is a transition from marine-grounded ice to ocean because much of the ice load will be replaced by water load. Have you plotted the net surface mass change, and would it be useful to include such plots to aid interpretation of the results? (ii) Errors peak along the location of the final grounding line in experiments ICE-GOL and ICE-RD (fig. 4). This is described as the 'load edge' at various points in the text but if I have correctly interpreted how net surface mass change is computed the grounding line will not necessarily align with the 'edge' of the load. Did you consider whether the misfit along the final grounding line position may be fundamentally related to representation of this transition within the GIA model? (iii) To understand how you calculate the 'Mass Factor' (fig. 3) a little more information is needed on how the model applies the load. Using the sensitivity experiments as an example; is the same load (i.e. 100m ice) applied to all elements that lie within, or partially within, the footprint of the load, or is the load scaled according to how much of each element is covered by the footprint of the cylinder?

Minor clarifications

Place names: rather than 'the West Antarctica' we tend to refer to 'West Antarctica' or 'the West Antarctic Ice Sheet' (similarly for East Antarctica), however, we do refer to 'the Antarctic Peninsula'. 'Central Antarctica' (line 218) is not standard terminology.

Section 2.2: I think the 15 km grid is used to produce results for section 4 but not section 3; it would be useful to clarify this somewhere in this section.

Line 162: 'incrementally smaller regions' – please clarify whether you use a series of nested regions, i.e. the 3.75km grid is always located within a slightly larger 7.5km grid, which is located within the 15km grid, or whether each higher-resolution grid is inserted directly into the 15km grid.

Line 164: 'a few layers down to 10 km' – line 158 states that the shallowest layer in the grid is at 12km; do you add extra layers when creating the higher-resolution grids?

Line 170: is the elastic/density structure used in the purely elastic model the same as that used in the 3D models (described on line 174)? What is the lithosphere thickness used in the elastic model?

Line 203: 'close to the preferred value in Kaufmann et al.' – I couldn't work out what this refers to since there is no mention of a scaling factor in the cited article. In general, it is not clear how the scaling factor is applied, making it difficult to derive useful insight from the values quoted here.

Lines 209-210: suggest '…viscosity estimates derived from GPS bedrock uplift rates in three regions' (note that the Antarctic Peninsula is often defined to be a separate region to the WAIS)

Line 227: is initial bedrock topography (as well as ice thickness) derived from Bedmap2?

Section 3.1/figs 3, 4, 7: when calculating the RMSE/percentage error in fig. 3, and the misfits shown in figs 4 and 7, how do you quantify the difference between results determined using different grids, i.e. in situations where output is produced at different resolutions?

Line 320: I think the ICE-SH model covers 25 years, between 1992-2017 (also check caption to fig. 2).

Line 373: should include references to fig. 2, not fig. 3?

Line 400-401: 'Along the final grounding line…' – based on fig. 7c, I don't think this statement holds.

Line 403: you mention five earth models, but only four are described in this paragraph. I suspect the fifth model is the 1D/WAIS model described in the next section, but this is not clear.

Line 407: I think the EM1_L and EM1_M models are derived using different 1D viscosity profiles as well as different viscosity scaling factors – this may be worth mentioning.

Lines 437-444: statements about time in this paragraph (e.g. 'after 50 years', 'within decades') are ambiguous because it is not clear whether they are referenced to the start of the model run (in 1950) or the start of the 21$^{st}$ century (as suggested by the opening sentence).

Line 452: is the 'maximum' value defined by the 75$^{th}$ percentile *plus* 1.5 times the interquartile range (I think 'minus' is correct for the 'minimum' value)? Comment also applies to various figure captions.

Line 469: 'the difference in predictions associated with earth model configuration lies between ~2-10% ...' – this statement is based on a comparison between results derived using the EM1_L and EM1_M models, which are derived from the same underlying seismic model, i.e., this is not the most extreme comparison that could be made. This statement may require a caveat.

Line 482: 'the elastic signal' -> 'the total signal'?

Line 539: a value of 7 +/- 3 is quoted elsewhere (e.g. line 300, caption to figure 3) – which is correct?

Figures: please add latitude and longitude labels to relevant figures.

Figures 1c,d and S1: I think the black line marks the edge of the ice shelf and the grey line marks the grounding line – not the 0 m contour, which is typically much further inland.

Figure 3: (i) 'calculated within 2 km of the loaded region' – does this statement apply to multiple plots in the figure? (ii) The ratio between cylinder radius and grid resolution is quoted as 3:1, not 1:3, in the main text (lines 298-299). (iii) Do the colours saturate in plots b-e?

Figure 5: (i) the reference to plot (a) is not needed. (ii) The reference to 'left to right' when describing what is represented by the edges of the boxes does not make sense – plot rotated?

Figure 8: (i) in the lower panel of fig. 8b, why does ice thickness not reach zero by 2100? My understanding is that the grounding line retreats upstream of this point during the model run. (ii) I suggest splitting the sentence that describes what is shown in plot (b) – it is very long! (iii) Please refer to Powell (2021) rather than the article in review.

Figure 9: please clarify which earth models were used to produce each set (row) of results.

Figure S1: please include a label indicating the depth associated with each plot in panels (a) and (b).

There are occasional small words missing or grammatical errors – please carry out a final check for such issues before resubmitting the article. Please also check that all values quoted in the text agree with values listed in the figures (e.g. line 344).

---

## Author Response (AR3)

(Author Response in Blue) We would like to thank the editor for taking the time to share these additional suggestion which have once again helped us to improve the manuscript. We have made the requested adjustments and hope they help provide the necessary clarification.

**Editor comments on "Resolving GIA in response to modern and future ice loss at marine grounding lines in West Antarctica" by Wan et al.**

I would like to thank the authors for the care they have taken in addressing my previous comments, particularly in relation to the terminology used to describe various processes associated with glacial isostatic adjustment and the key result that relates grid resolution to load radius. All other points were addressed in the authors' rebuttal and only a few minor queries remain. These are listed below (line numbers refer to the 'clean' manuscript version 4), and I am delighted to confirm that this article is now accepted for publication (subject to technical corrections, i.e. no further review).

Thank you for choosing to publish your work in The Cryosphere. Pippa Whitehouse (editor)

Line 171: Line 164 states that the shallowest layer is at 12 km depth so it is not clear how there can be 'a few layers down to ~10 km depth'. Do you include extra layers when you carry out the grid refinement?

We have clarified that the 3-D region of grid refinement is performed to depths of ~ 10 km, and operates by bisecting the grid both horizontally and vertically.

Lines 191-193: text that describes how you use the free scaling parameter is difficult to understand without close reading of the supplementary material. For example, it is not clear what you mean by 'the depth dependence' and 'the surface value'. Please simplify this text and explain the meaning of '1/C' (appears after the scaling factors in the paragraph beginning line 211)

We have removed the concept of depth dependence and surface value to minimise confusion. 1/C is just the dimension / unit of the scaling factor – we've changed it to $K^{-1}$ for better clarity.

Line 394: inconsistent use of a minus sign when reporting the ICE-RD result compared with the ICE-SH and ICE-GOL results

Thank you for spotting this inconsistency – we have removed the minus sign.

Lines 435-446: this paragraph was written before you included the 1-D (WAIS) simulations (which are introduced in the previous paragraph and discussed in the next paragraph). This leads to some ambiguity here because it is not clear what you mean by 'the average 1-D viscoelastic Earth model' – all 1-D models can essentially be described as 'average' models, and results for two different 1-D models are shown in fig. 8, which is discussed in this paragraph. Please use a consistent naming convention for the two 1-D models used in this study.

Thank you for bringing up this area of confusion. We have clarified which model is the 1-D global average vs. 1-D WAIS best-fit earth model wherever referred to.

Lines 501-502: I am confused by the statement that relates the viscous and elastic components of the signal (apologies if I am being dense!). I see that the percentages are taken from fig. 9 (bottom row), where the x-axis is labelled '% of total signal'. This suggests that the values quoted here describe the relative magnitude of (a) the difference between the elastic and viscous components and (b) the total (viscoelastic?) signal; this is different to what is implied by your text (the relative

magnitude of the elastic and viscous signals). Please clarify what you mean by the 'total signal' and hence check the wording of this result.

Thank you for catching this as well, we have clarified in the text and in the caption for Fig. 9 as to which experimental sea level result the % is referring to.

Caption to Fig. 1 (and Fig. S1): The grounding line is not the same as the 0m contour. In fact, the grounding line delineates the extent of marine-grounded ice (which is currently mentioned in relation to the black line, which delineates the extent of floating ice)

We have removed the mention of the 0m contour.

Caption to Fig. 9: I think 8 factors are compared in this figure, not 6 (labelled on the left)

Thank you for catching this, we have clarified the caption accordingly.